EMBO
Molecular Medicine

# Inhibition of transcription by dactinomycin reveals a new characteristic of immunogenic cell stress

Juliette Humeau[1,2,3], Allan Sauvat[1,2], Giulia Cerrato[1,2,3], Wei Xie[1,2,3], Friedemann Loos[1,2] (iD),
Francesca Iannantuoni[1,2,4], Lucillia Bezu[1,2,4], Sarah Lévesque[1,2,3], Juliette Paillet[1,2,3], Jonathan Pol[1,2] (iD),
Marion Leduc[1,2], Laurence Zitvogel[3,5,6,7], Hugues de Thé[8,9], Oliver Kepp[1,2,*] (iD) &
Guido Kroemer[1,2,10,11,12,**] (iD)

## Abstract

Chemotherapy still constitutes the standard of care for the treatment of most neoplastic diseases. Certain chemotherapeutics from the oncological armamentarium are able to trigger *pre-mortem* stress signals that lead to immunogenic cell death (ICD), thus inducing an antitumor immune response and mediating long-term tumor growth reduction. Here, we used an established model, built on artificial intelligence to identify, among a library of 50,000 compounds, anticancer agents that, based on their molecular descriptors, were predicted to induce ICD. This algorithm led us to the identification of dactinomycin (DACT, best known as actinomycin D), a highly potent cytotoxicant and ICD inducer that mediates immune-dependent anticancer effects *in vivo*. Since DACT is commonly used as an inhibitor of DNA to RNA transcription, we investigated whether other experimentally established or algorithm-selected, clinically employed ICD inducers would share this characteristic. As a common leitmotif, a panel of pharmacological ICD stimulators inhibited transcription and secondarily translation. These results establish the inhibition of RNA synthesis as an initial event for ICD induction.

**Keywords** dactinomycin; eIF2α phosphorylation; immunogenic cell death; transcription; translation
**Subject Categories** Cancer; Computational Biology; Pharmacology & Drug Discovery

## Introduction

The last decade has witnessed the clinical implementation of anti-cancer immunotherapies (Nishino *et al*, 2017), as well as the realization that the long-term success of other antineoplastic therapies (i.e., with cytotoxicants, irradiation or targeted agents) beyond therapy discontinuation depends on the reinstatement of immuno-surveillance (Vesely *et al*, 2011). Indeed the density, composition, and functional state of the tumor immune infiltrate, the "immune contexture" has a decisive impact on the outcome of such non-immune targeted therapies (Fridman *et al*, 2017). Successful chemotherapeutic agents (Obeid *et al*, 2007b; Tesniere *et al*, 2010), radiotherapy (Golden *et al*, 2012), and some targeted agents (Menger *et al*, 2012; Liu *et al*, 2019) kill cancer cells in a way that they become recognizable by the immune system, hence causing "immunogenic cell death" (ICD) (Casares *et al*, 2005; Galluzzi *et al*, 2017). As a general rule, immunogenicity results from the combination of two phenomena, namely (i) antigenicity, implying that tumor cells must be antigenically distinct from their normal counterparts, and (ii) adjuvanticity, meaning that stressed and dying neoplastic cells must emit damage-associated molecular patterns (DAMPs) to activate innate immune effectors (Galluzzi *et al*, 2017). Although cancer therapies may affect the immunopeptidome presented by class I molecules at the cancer cell surface (Bloy *et al*, 2017), the most important effect of ICD concerns the release/exposure of DAMPs (Kroemer *et al*, 2013).

At the molecular level, ICD is characterized by an autocrine stimulation of type-1 interferon (IFN) receptors (Sistigu *et al*, 2014), the pre-apoptotic exposure of calreticulin (CALR) on the cell surface

1 Equipe labellisée par la Ligue contre le Cancer, Sorbonne Université, INSERM UMR1138, Centre de Recherche des Cordeliers, Université de Paris, Paris, France
2 Metabolomics and Cell Biology Platforms, Gustave Roussy, Villejuif, France
3 Faculty of Medicine Kremlin Bicêtre, Université Paris Sud, Paris Saclay, Paris, France
4 Hospital Doctor Peset - FISABIO, Valencia, Spain
5 Gustave Roussy Cancer Campus (GRCC), Villejuif, France
6 INSERM U1015, Villejuif, France
7 Center of Clinical Investigations in Biotherapies of Cancer (CICBT), Villejuif, France
8 College de France, INSERM UMR 1050, CNRS UMR 7241, PSL University, Paris, France
9 INSERM UMR 944, CNRS UMR 7212, Equipe labellisée par la Ligue contre le Cancer, IRSL, Hopital St. Louis, Université de Paris, Paris, France
10 Suzhou Institute for Systems Medicine, Chinese Academy of Medical Sciences, Suzhou, China
11 Pôle de Biologie, Hôpital Européen Georges Pompidou, AP-HP, Paris, France
12 Department of Women's and Children's Health, Karolinska Institutet, Karolinska University Hospital, Stockholm, Sweden
*Corresponding author. Tel: +33 1 42 11 45 16; E-mail: captain.olsen@gmail.com
**Corresponding author. Tel: +33 1 44 27 76 67; E-mail: kroemer@orange.fr

(Obeid *et al*, 2007a,b; Panaretakis *et al*, 2008), the release of ATP during the blebbing phase of apoptosis (Martins *et al*, 2014), the post-apoptotic/post-necrotic exodus of annexin A1 (ANXA1) (Vacchelli *et al*, 2015; Baracco *et al*, 2016), and chromatin-binding protein high mobility group B1 (HMGB1) (Apetoh *et al*, 2007). Type-1 interferon secretion depends on the stimulation of several pattern recognition receptors (PRRs including TLR3 and cGAS/STING) (Sistigu *et al*, 2014), CALR exposure on a partial endoplasmic reticulum stress response (Panaretakis *et al*, 2009; Bezu *et al*, 2018), ATP release on pre-mortem autophagy (Michaud *et al*, 2011; Martins *et al*, 2012), and ANXA1/HMGB1 exodus on secondary necrosis (Apetoh *et al*, 2007; Vacchelli *et al*, 2015). ATP, ANXA1, CALR, and HMGB1 interact with four receptor types, namely, purinergic P2Y2 or P2X7 receptors (Ghiringhelli *et al*, 2009), formyl peptide receptor-1 (FPR1) (Vacchelli *et al*, 2015), CD91 (Garg *et al*, 2012), and toll-like receptor 4 (TLR4) (Apetoh *et al*, 2007; Yamazaki *et al*, 2014), respectively, that are present on the surface of dendritic cells (DCs) or their precursors. P2RY2/P2RX7, FPR1, CD91, and TLR4 promote chemotaxis, juxtaposition with dying cells (Vacchelli *et al*, 2015), subsequent engulfment of portions of dying cells (Obeid *et al*, 2007b) in addition to production of interleukin-1β (Sistigu *et al*, 2014), and cross-presentation of tumor antigens (Ma *et al*, 2013), by DCs, respectively.

Of note, inhibition of the aforementioned ligand–receptor interactions abolishes the efficacy of anticancer ICD-inducing therapies in pre-clinical models (Apetoh *et al*, 2007; Ghiringhelli *et al*, 2009; Garg *et al*, 2012; Vacchelli *et al*, 2015). Moreover, there is an abundant literature suggesting that deficiencies in these ligand and/or receptors (and their downstream signals) have a negative impact on patient prognosis, predicting therapeutic failure (Apetoh *et al*, 2007; Ghiringhelli *et al*, 2009; Vacchelli *et al*, 2015). In patients, suboptimal therapeutic regimens (failing to induce ICD) (Tesniere *et al*, 2010; Pfirschke *et al*, 2016), selective alterations in cancer cells (preventing the emission of immunogenic signals during ICD), and inherited or acquired defects in immune effectors (abolishing the perception of ICD by the immune system) can contribute to therapeutic failure due to an insufficient immune recognition of malignant cells (Kroemer *et al*, 2013). Importantly, ICD induction can synergize with subsequent immune checkpoint blockade targeting PD-1/PD-L1 interaction (Pfirschke *et al*, 2016; Liu *et al*, 2019) and hundreds of clinical trials are investigating such combination effects (www.clinicaltrials.gov).

Given the rising significance of ICD, it is important to understand the rules governing its induction at the cellular and molecular levels, especially considering that their comprehension may facilitate the identification of novel, effective ICD inducers. Here, we report the discovery of dactinomycin (DACT, commonly known as actinomycin D), a chemotherapeutic agent used to treat various sarcomas and an efficient inhibitor of transcription, as an ICD inducer. Based on this serendipitous finding, we developed the concept that inhibition of RNA synthesis is a close-to-common feature of ICD.

# Results

## Identification of dactinomycin as a *bona fide* ICD inducer

We used an artificial intelligence machine learning approach (Bezu *et al*, 2018) to predict the probability of inducing ICD of 50,000

distinct compounds tested for their anticancer effects on the NCI-60 panel of human tumor cell lines (Shoemaker, 2006; Fig 1A), while plotting the ICD prediction score against their mean $IC_{50}$, *i.e.*, the dose that reduces cell proliferation by half (Fig 1B). The compounds that exhibited cytotoxicity and an ICD score higher than mitoxantrone (MTX), a standard ICD inducer (Obeid *et al*, 2007b; Ma *et al*, 2011), were considered as potential ICD inducers. Two compounds, among the ones that have entered clinical trials, stood out as drugs having a low $IC_{50}$ and a high ICD score. Trabectedin is known for its capacity to selectively eliminate tumor-associated macrophages, which explains at least part of its anticancer activity (Germano *et al*, 2013). Dactinomycin (DACT, best known as actinomycin D, a product of *Streptomyces parvulus*), which is generally considered as a DNA intercalator that inhibits topoisomerases and RNA polymerases (Goldberg *et al*, 1962), is used for the treatment of childhood-associated sarcomas (Wilms, Ewing, rhabdomyosarcoma), gestational trophoblastic disease including hydatidiform moles and choriocarcinomas (Khatua *et al*, 2004; Turan *et al*, 2006), and some types of testicular cancers (Early & Albert, 1976). We therefore evaluated DACT for its capacity to induce ICD.

When added to human osteosarcoma U2OS cells engineered to express a CALR-green fluorescent protein (GFP) fusion protein, DACT (used around the $IC_{60}$ for these cells, *i.e.*, at 0.5 and 1 μM, Appendix Fig S1) caused peripheralization of the green fluorescence to the same extent as the positive control, MTX, as determined by videomicroscopy (Fig 2A–C). Similarly, live-cell imaging revealed the decrease of HMGB1-GFP in the nuclei of DACT-treated cells (Fig 2D–F). DACT also reduced the ATP-dependent quinacrine fluorescence staining of cells (Fig 2G and H), and the supernatants of DACT-treated cells stimulated the expression of MX1, a type 1 interferon-related biosensor, with GFP under the control of its promoter (Fig 2I and J). Alternative methods were then used to measure the emission of endogenous DAMPs. Thus, the plasma membrane surface exposure of CALR on viable cells was detected by flow cytometry (Fig 2K and L); the release of endogenous HMGB1 into the culture medium was confirmed by ELISA (Fig 2M), and ATP release into the supernatant of DACT-treated cells was assessed by a luciferin conversion assay (Fig 2N).

One of the pathognomonic features of ICD is a partial endoplasmic reticulum (ER) stress response that involves phosphorylation of eukaryotic initiation factor 2α (eIF2α) without activation of its downstream factor ATF4, and without the ATF6 and the IRE1/XBP1 arms of the unfolded protein response (Panaretakis *et al*, 2009; Pozzi *et al*, 2016; Bezu *et al*, 2018). Accordingly, DACT caused eIF2α phosphorylation (measured by immunofluorescence, Fig 3A and B), but neither significant downstream ATF4 activation (expressed as a GFP fusion protein, Fig 3C and D), nor ATF6 translocation from the cytosol to the Golgi and to nuclei (detected as a GFP fusion protein, Fig 3E and F), nor expression of an XBP1-(DBD-venus fusion protein that is only in-frame for venus (a variant of GFP) when XBP1 has been spliced by IRE1 (Fig 3G and H). We knocked out each of the four eIF2α kinases (EIF2AK1 to 4) in U2OS cells (Appendix Fig S2) and determined their contribution to DACT-induced eIF2α phosphorylation. As for other ICD inducers (such as anthracyclines and oxaliplatin) (Panaretakis *et al*, 2009), EIF2AK3 (also known as PERK) was responsible for DACT-stimulated eIF2α phosphorylation (Fig 3I and J). Human U2OS osteosarcoma cells expressing a CALR-RFP fusion protein were implanted in immunodeficient mice

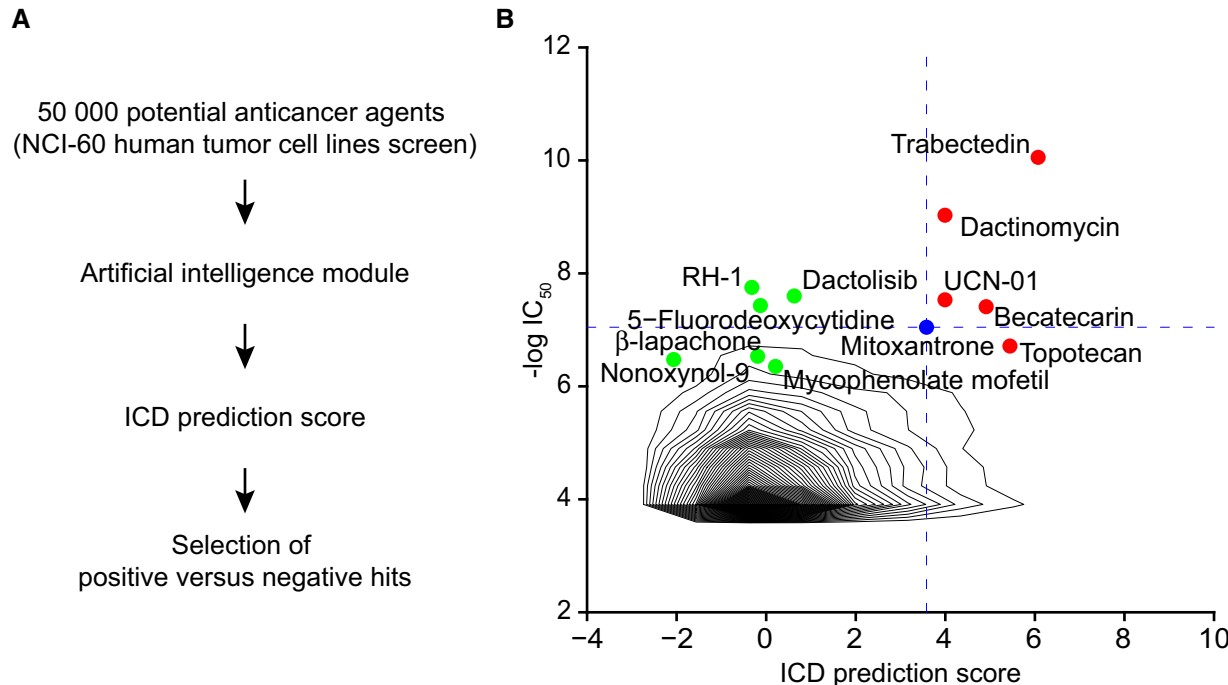

**Figure 1. Prediction of immunogenic cell death (ICD).**

A  The 50,000 potential anticancer agents from the NCI-60 human tumor cell lines screen were analyzed with an artificial intelligence model that can predict immunogenic cell death (ICD) based on molecular descriptors.

B  The distribution of the drugs based on their $IC_{50}$ and predicted ICD score is depicted as density plot. Based on the properties of the standard ICD inducer mitoxantrone (blue), we selected negative (green) and positive (red) hits: Agents that entered into clinical trials, having an $IC_{50} < 1$ μM (and therefore $-\log(IC_{50}) > 6$) and whose predicted ICD score is higher than the ICD score of mitoxantrone, are potential ICD inducers (positive hits). Some agents that entered into clinical trials, whose $IC_{50} > 1$ μM and which have an ICD prediction score lower than 1, are negative hits.

to generate tumors that were then treated with DACT. DACT induced the rapid (6 h) phosphorylation of eIF2α (Fig 3K and L) and the redistribution of CALR-RFP to the cell periphery within 24 h (Fig 3M and N). DACT also stimulated all ICD hallmarks (eIF2α phosphorylation, CALR exposure, ATP and HMGB1 release, as well as induction of type 1 interferon-related metagene) in another cell line, the murine methylcholanthrene-induced fibrosarcoma MCA205 (Fig EV1). Altogether, these results confirm the capacity of DACT to stimulate a signal transduction pathway that leads to immunogenic stress and death.

**Immune-dependent anticancer effects of dactinomycin**

To further investigate the immunogenic properties of DACT, we treated cancer cells *in vitro* with this compound, labeled them with CellTracker orange (CMTMR) and measured their engulfment by bone marrow-derived CD11c⁺ dendritic cells (BMDCs) (Fig 4A). DACT was able to stimulate the temperature-dependent phagocytosis by BMDCs (Fig 4B, Appendix Fig S3). Moreover, BMDCs exposed to DACT-treated tumor cells upregulated MHC class II and the co-stimulatory molecule and activation marker CD86 (Fig 4A, C and D, Appendix Fig S4). Next, we determined the capacity of DACT to induce ICD in vaccination assays. For this, MCA205 cells were treated with the cytotoxicants *in vitro*, washed, and then injected subcutaneously (*s.c.*) in the absence of any adjuvant into immunocompetent

C57Bl/6 mice. These animals were then rechallenged 2 weeks later with living MCA205 cells injected into the opposite flank (Fig 4E). As compared to controls, mice vaccinated with DACT-treated cells exhibited a delay in tumor growth (Fig 4F and G).

Next, we administered DACT alone or in combination with the widely used and non-immunogenic chemotherapeutic cis-dichloro-diammine-platinum (cisplatin; CDDP) (Casares *et al*, 2005; Martins *et al*, 2011) and an anti-PD-1 antibody, to immunocompetent C57Bl/6 mice bearing MCA205 (Fig EV2A and K). While DACT alone was not able to significantly reduce tumor growth, the combination of CDDP and DACT was efficient against established MCA205, reducing tumor growth and extending overall survival (Fig EV2A–E). However, this antineoplastic effect of CDDP plus DACT was lost when the cancer cells were growing in immunodeficient *nu/nu* mice that are athymic and hence lack mature T lymphocytes (Fig EV2F–J). Of note, DACT followed by PD-1 blockade delayed tumor growth and the triple combination (CDDP+DACT+ PD-1 blockade) allowed for the permanent cure of 3 out of 9 MCA205 cancers bearing mice (Fig EV2K–O). When cured mice were re-inoculated with MCA205 cancers, no tumor growth was observed, although antigenically unrelated TC-1 non-small-cell lung cancer readily formed macroscopic cancers in these animals (Fig EV2P and Q). Hence, in combination regimens, DACT can be used to stimulate a curative anticancer immune response that generates protective long-term memory.

In accordance with the hypothesis that DACT mediates immunostimulatory effects *in vivo*, the immune infiltrate of tumors from mice receiving systemic DACT exhibited an improved ratio of CD8[+] T lymphocytes over CD4[+]FoxP3[+] regulatory cells as well as an increase in the percentage of NK and NKT cells (Fig EV3B–D). Following a non-specific restimulation with phorbol myristate acetate (PMA) and ionomycin, an augmentation in IL17-producing CD4[+], CD8[+], and γδ T cells was observed in the tumor infiltrate (Fig EV3D–F). In addition, DACT tended to enhance the secretion of IFNγ by CD8[+] T cells (Fig EV3E and F), while the amount of IL4 produced by CD4[+] T cells remained unchanged (Fig EV3I).

The immunostimulatory effect of DACT was recapitulated in WEHI 164 cells, yet another methylcholanthrene-induced fibrosarcoma, alone and in combination with anti-PD-1 checkpoint

blockade (Fig 5A). In this model, DACT treatment alone sufficed to cure 5 out of 8 mice from transplanted fibrosarcoma (Fig 5B), an effect which was completely abolished when CD4[+] and CD8[+] T cells were depleted with specific antibodies (Fig 5C–E). DACT combined with anti-PD-1 checkpoint blockade led to the cure or disease control in all treated animals (Fig 5F–H). Mice cured from their sarcoma by DACT-based chemotherapy or anti-PD-1/DACT-based immunochemotherapy were rechallenged with WEHI 164 in the opposite flank. All 11 mice remained tumor-free after several months, revealing a protective immune memory response. Of note, WEHI 164 tumors from DACT-treated mice exhibited an increased expression of the mRNA coding for IFNγ, as determined by quantitative reverse transcription polymerase chain reaction (qRT–PCR) (Fig 6A and B). The role of IFNγ in the anticancer effect of DACT

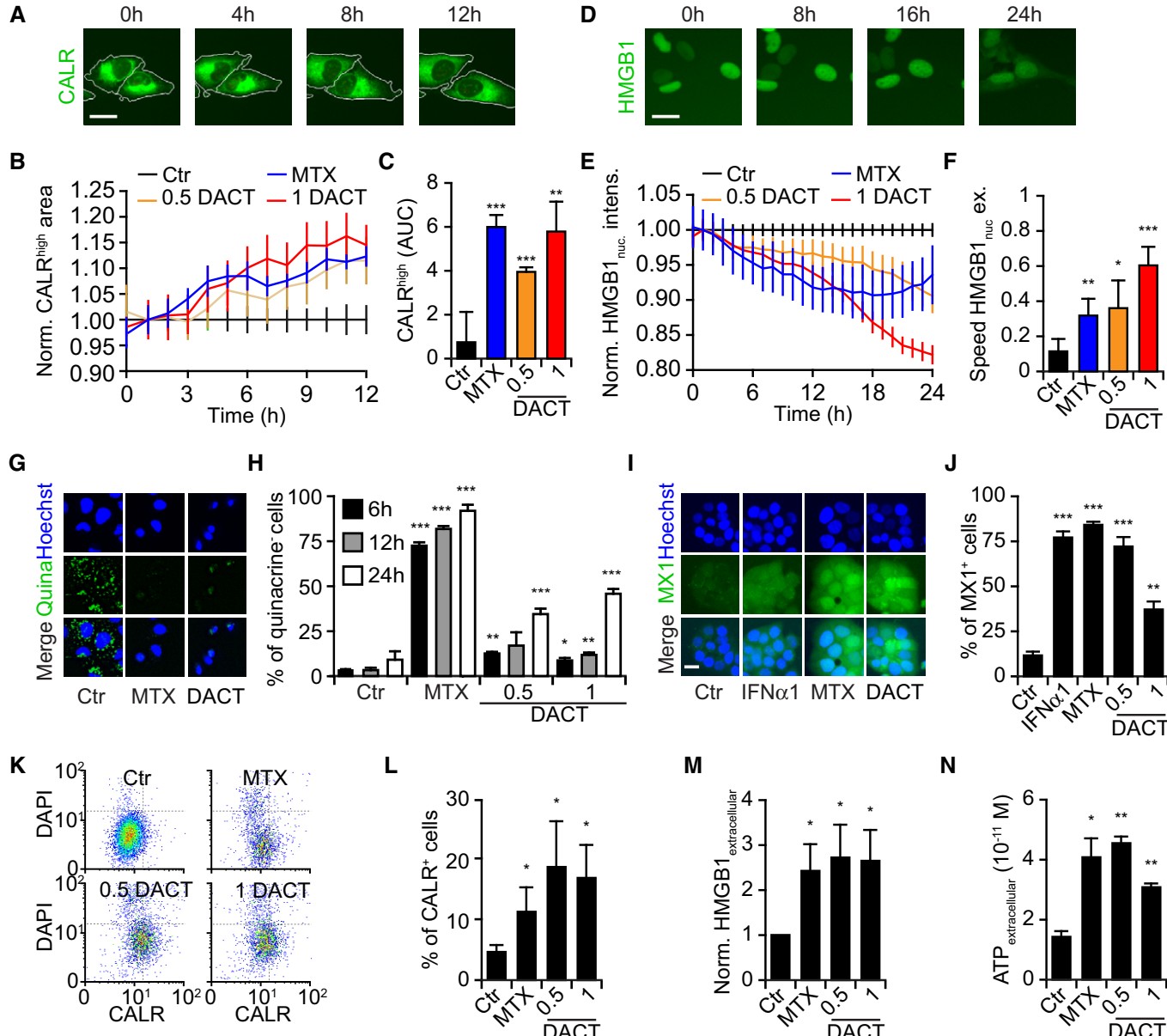

Figure 2.

**Figure 2. ICD hallmarks in human cancer cells.**

Human osteosarcoma U2OS cells were treated with dactinomycin (DACT) at 0.5 or 1 μM, or with mitoxantrone (MTX) between 1 and 6 μM as positive control (A-N).

A–C Human osteosarcoma U2OS cells stably expressing CALR-GFP and H2B-RFP were treated as described above, and images were acquired once per hour for 12 h (A). For one representative experiment among three, the mean ± SEM of the average area of high CALR dots (normalized to the control at each time point) of quadruplicates is shown (B). Values are depicted as the area under the curve mean ± SD of triplicates (C).

D–F Treated U2OS cells stably expressing HMGB1-GFP and H2B-RFP images were acquired every hour for 24 h (D). For one representative experiment among three, the mean ± SEM of the green fluorescence intensity in the nucleus (normalized to the control at each time point) of quadruplicates is depicted (E). For each cell, the speed of nuclear release (difference of HMGB1 nuclear green fluorescence intensity between two time points) was calculated. Values are depicted as the average speed of the nuclear release mean ± SD of quadruplicates (F).

G, H U2OS cells were treated for 6, 12, or 24 h, and ATP was stained with quinacrine (G). The number of quinacrine negative cells was assessed based on the distribution of cellular green fluorescence intensity in MTX versus control conditions. For one representative experiment among three, the mean ± SD of quadruplicate assessments is shown (H).

I, J U2OS wild-type cells were treated with MTX or DACT as described above for 6 h. Then, medium was refreshed and 24 h later, type I interferon response was assessed by transferring the supernatant on HT29 MX1-GFP reporter cells lines cells for additional 48 h. Human type 1α interferon (IFNα1) was also added on the cells as an additional positive control. Images were acquired by fluorescence microscopy, and the number of positive cells was assessed based on the distribution of cellular green fluorescence intensity in IFNα1 versus control conditions (I). The percentage of MX1-positive cells was calculated, and the mean ± SEM of five independent experiments is depicted (J).

K, L U2OS wild-type cells were treated as mentioned above for 6 h, and then, medium was refreshed. Twenty-four hours later, cells were collected and surface-exposed calreticulin (CALR) was stained with an antibody specific for CALR. DAPI was used as an exclusion dye, and cells were acquired by flow cytometry (K). The percentage of CALR$^+$ cells among viable (DAPI$^-$) ones is depicted. The mean ± SEM of six independent experiments is depicted (L).

M U2OS cells were treated as described above for 24 h, and the concentration of HMGB1 released in the supernatant was quantified with an ELISA kit and then normalized to control. The mean ± SEM of four independent experiments is shown.

N U2OS were treated as described above for 24 h. Concentration of secreted ATP in the supernatant was quantified with a luciferase-based bioluminescence kit. The mean ± SD of quadruplicates from one representative among three experiments is depicted.

Data information: Scale bars represent 20 μm. All *P*-values showing significances of treatments compared to control (Ctr) were calculated with Student's *t*-test: **P* < 0.05, ***P* < 0.01, ****P* < 0.001.

---

was confirmed in this model as injections of IFNγ blocking antibody reduced the therapeutic effect of DACT (Fig 6C–F).

**Inhibition of transcription by a panel of ICD inducers**

DACT is known to suppress the transcription of DNA to RNA (Goldberg *et al*, 1962, 1963; Bensaude, 2011). Accordingly, DACT reduced the incorporation of the mRNA precursor 5-ethynyl uridine (EU) into cells, as revealed by means of click biochemistry yielding a fluorescent signal (Jao & Salic, 2008). Astonishingly, a series of other established ICD inducers also inhibited transcription, as documented for the anthracyclines daunorubicin, doxorubicin, epirubicin, and mitoxantrone (Casares *et al*, 2005; Obeid *et al*, 2007b), oxaliplatin (Tesniere *et al*, 2010), and crizotinib (Liu *et al*, 2019). Bortezomib, which has been identified as an ICD inducer on myeloma cells (Spisek *et al*, 2007; Garg *et al*, 2017), had a rather partial effect and only at high doses. Several microtubular inhibitors (docetaxel, paclitaxel, vinblastine, and vincristine) which induce CALR exposure yet have not been reported to induce ICD *in vivo* (Alagkiozidis *et al*, 2011; Senovilla

---

**Figure 3. DACT induces a split ER stress response in U2OS.**

A, B Human osteosarcoma U2OS cells were treated with different concentrations of dactinomycin (DACT) (0.25, 0.5 or 1 μM) for 6 h. Thapsigargin (THAPS) at 3 μM was used as a positive control. After fixation, cells were stained with phospho-eIF2α (Ser51)-specific antibody followed by an Alexa Fluor-647 secondary antibody, nuclei were counterstained with Hoechst 33342, and phosphorylation was assessed by fluorescence microscopy. Images were segmented and analyzed, and the red cytoplasmic fluorescence intensity was measured.

C, D U2OS cells stably expressing ATF4-reporter were treated as described above for 12 h. The expression and nuclear translocation of ATF4 were assessed by fluorescence microscopy, and the nuclear green fluorescence intensity was quantified.

E, F U2OS cells stably expressing ATF6-GFP were treated as described above, and nuclear translocation of ATF6 was represented as the ratio of nuclear versus cytoplasmic green fluorescence intensity.

G, H U2OS cells stably expressing venus in-frame with alternatively spliced XBP1 (sXBP1) were treated as described above for 12 h. *De novo* expressed venus was measured intracellular.

I, J U2OS wild-type and knock-out for eIF2α kinases 1, 2, 3, and 4 cells were treated with 3 μM THAPS as a positive control for EIF2AK3-mediated eIF2α phosphorylation or with 1 μM DACT for 6 h. After fixation, cells were stained for peIF2α as described above and cytoplasmic intensity was quantified.

K–N U2OS CALR-RFP cells were inoculated subcutaneously (*s.c*) in the flank of *nu/nu* mice. Tumors were injected with PBS (Ctr) (*n* = 5), 0.5 mg/kg tunicamycin (TM) for 6 h (*n* = 3) or 0.5 mg/kg DACT for 6 h (*n* = 4) or 24 h (*n* = 3). Tumors were cut and stained with a phospho-eIF2α (Ser51)-specific antibody followed by an Alexa Fluor-647 secondary antibody and counterstained with Hoechst 33342. Two slices per tumor were imaged for their DAPI, RFP, and Cy5 signals. Out-of-focus images were removed from the dataset leading to the analysis of the following conditions: 8 × Ctr, 5 × TM, 8 × DACT 6 h, and 6 × for DACT 24 h. PeIF2α was quantified measuring Cy5 intensity in the cytoplasm (K, L) and CALR translocation by measuring the coefficient of variation (CV) of the RFP signal in the cytoplasm (M, N). Representative images of peIF2α are shown for Ctr, TM, and DACT at 6 h (K), whereas images of CALR are shown for Ctr and DACT at 24 h (M).

Data information: Images are shown for untreated control cells (Ctr), THAPS and DACT at 1 μM (A, C, E, G, I). Scale bars represent 20 μm (A, C, E, G, I) or 10 μm (K, M). For all barcharts, the mean ± SEM of three to five independent experiments is shown. Each value was compared to the control, and the *P*-value was calculated with Student's *t*-test: **P* < 0.05, ***P* < 0.01, ****P* < 0.001 (B, D, F, H). For peIF2α quantification, statistics were calculated using pairwise multiple comparison with a Benjamin–Hochberg correction. Stars indicate differences of each treatment compared to the control in the wild-type cells, whereas hashes indicate differences comparing the kinases knock-out cells to the wild-type cells for the same treatment: **/##*P* < 0.01, ***/###*P* < 0.001 (J). For *ex vivo* data, results are shown as dot plots with mean ± SD, and *P*-values were calculated using Student's *t*-test: **P* < 0.05, ***P* < 0.01 (L, N).

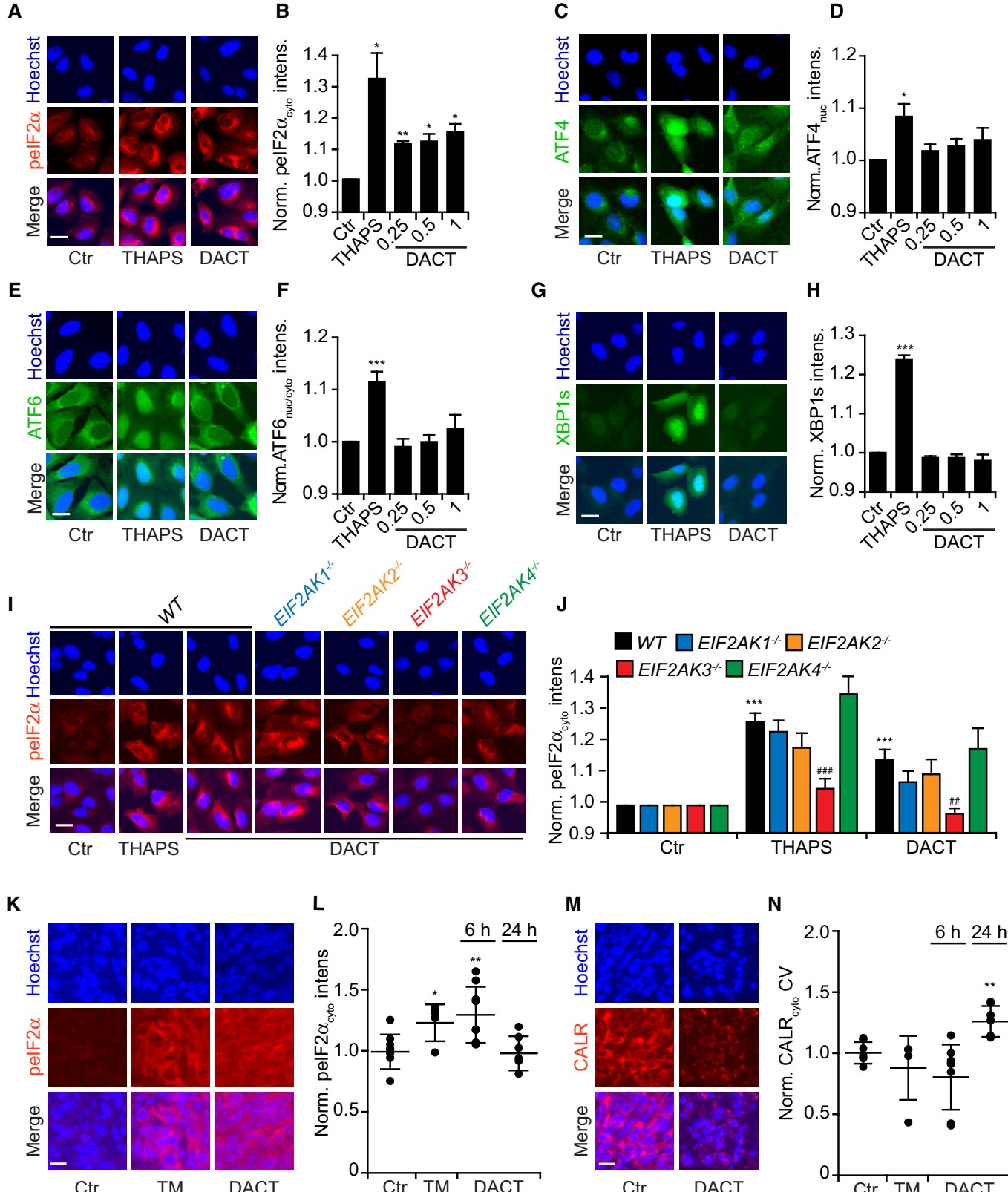

**Figure 3.**

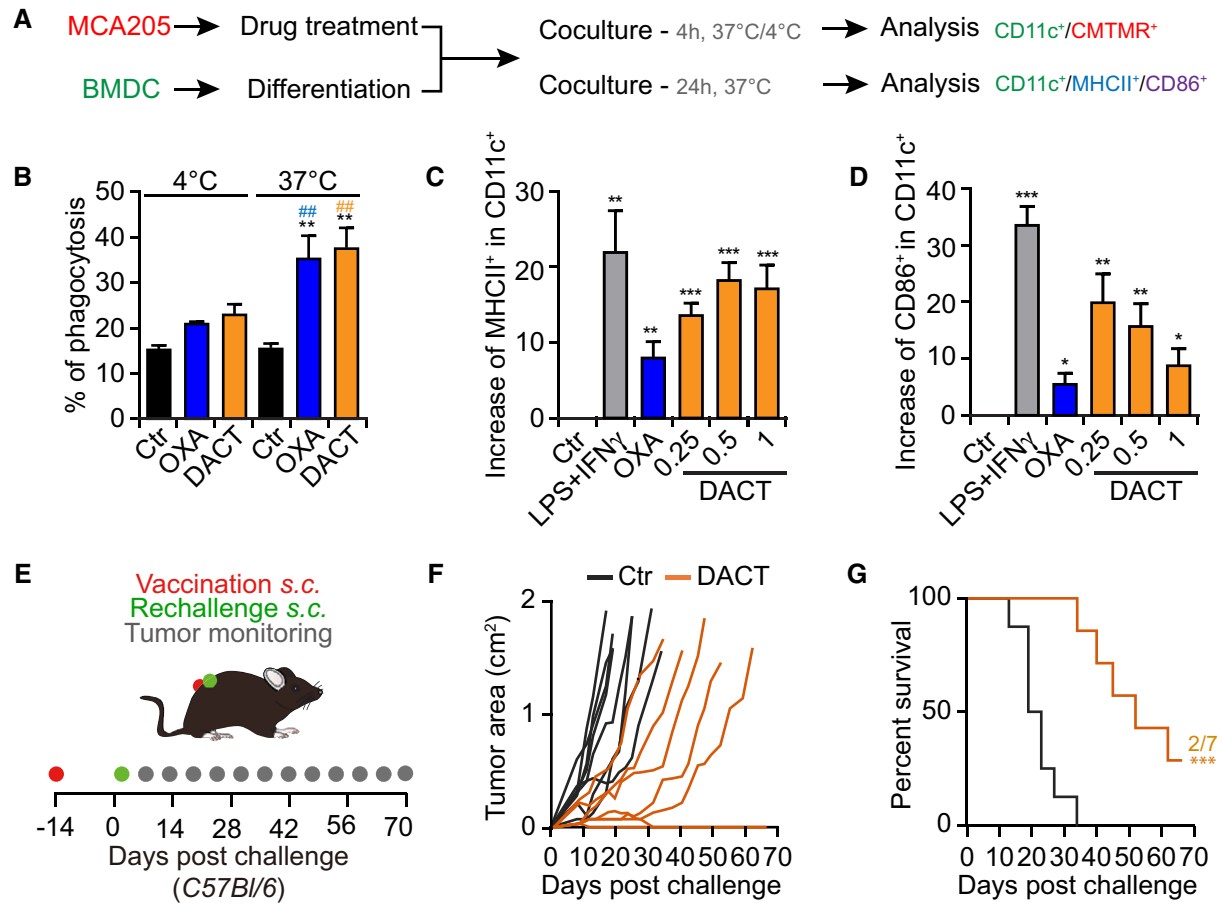

**Figure 4. DACT-treated cells activate the immune system.**

A, B   Mouse fibrosarcoma MCA205 cells were stained with CellTracker Orange (CMTMR) and treated for 24 h with 1 μM dactinomycin (DACT) or 500 μM oxaliplatin (OXA) as a positive control. Then, untreated or dying MCA205 cells were co-cultured with differentiated bone marrow-derived dendritic cells (BMDCs) for 4 h at 37 or at 4°C. Cells were collected, and dendritic cells were stained with CD11c-specific antibody before analysis by flow cytometry (A). The percentage of CMTMR and CD11c double-positive cells among all CD11c+ cells is indicated (B). The mean of three independent experiments ± SEM is depicted, and P-values were calculated using pairwise multiple comparisons test with a Benjamin–Hochberg correction. Stars indicate significant differences between each treatment and its corresponding control at the same temperature, whereas hashes indicate significant differences comparing the co-culture at 37°C to the co-culture at 4°C for the same treatment: **/##P < 0.01.

A, C, D   MCA205 cells were treated for 24 h with 500 μM oxaliplatin (OXA) or with 0.25, 0.5 or 1 μM dactinomycin (DACT). Then, untreated or dying MCA205 cells were co-cultured with BMDCs for 24 h at 37°C. As a positive control, BMDCs were co-cultured with 1 μg/ml LPS and 100 ng/ml IFNγ. Cells were collected and stained with a marker of viability, as well as with CD11c-, MHCII-, and CD86-specific conjugated antibodies, and analyzed by flow cytometry (A). The increase in the percentages of MHCII+ (C) and CD86+ (D) cells among CD11c+ cells was quantified with respect to untreated controls. The mean of five independent experiments ± SEM is depicted, and the P-values were calculated using Student's t-test: *P < 0.05, **P < 0.01, ***P < 0.001.

E–G   1 × 10^6 mouse fibrosarcoma MCA205 cells were treated in vitro with 1 μM dactinomycin (DACT). Dying cells were harvested and injected subcutaneously (s.c.) into one flank of immunocompetent syngeneic C57Bl/6 mice (n = 8 mice) to assess vaccination efficacy. PBS was injected in the control group (Ctr). Two weeks later, animals were rechallenged with 1 × 10^5 untreated MCA205 cells in the contralateral flank of the animals (E). Next, tumor size was measured regularly and individual tumor growths of DACT-vaccinated versus Ctr mice are depicted (F). Overall survival is depicted, and P-values (***P < 0.001.) were calculated with a log-rank test (G). One representative experiment among three is shown (F, G).

et al, 2012; Wang et al, 2015) had no effect on transcription. Cisplatin, a drug that is not considered as an efficient ICD inducer (Casares et al, 2005), had partial effects (Fig 7A and B). An alternative method for measuring stalled transcription consists in determining the separation of fibrillarin (a nucleolar marker) and nucleolin (which spreads from the nucleolus to the entire nucleus when rRNA synthesis is inhibited) by immunofluorescence (Peltonen et al, 2014; Sauvat et al, 2019). The profile of inhibition obtained using this assay was very similar to that obtained with

EU (Fig 7C and D). Inhibition of mRNA translation into proteins was also measured by monitoring the incorporation of the amino acid analogue L-azidohomoalanine (AHA) into cells (Wang et al, 2017). In this assay, all the tested agents caused an at least partial inhibition of protein synthesis. The magnitude of inhibition of translation correlated with that observed for transcription (Fig 7E and F). Both assays measuring transcription (Fig 7G) and translation (Fig 7H) correlated among each other underlining the validity of the obtained results.

Of note, this inhibition of translation was fully observable in cells homozygous for a non-phosphorylable eIF2α mutation (eIF2αS51A) generated by CRISPR-Cas9-mediated knock-in or when the downstream effects of peIF2α were abolished by integrated stress response inhibitor (ISRIB). As a positive control, the ER stress inducer thapsigargin, known to inhibit translation

downstream eIF2α phosphorylation (Sidrauski et al, 2013), did not induce the inhibition of translation in eIF2αS51A mutants or in ISRIB-treated cells (Fig EV4A). Hence, the ICD-induced inhibition of translation is likely secondary to the inhibition of transcription rather than a direct effect of EIF2AK3 on the translation-relevant factor eIF2α.

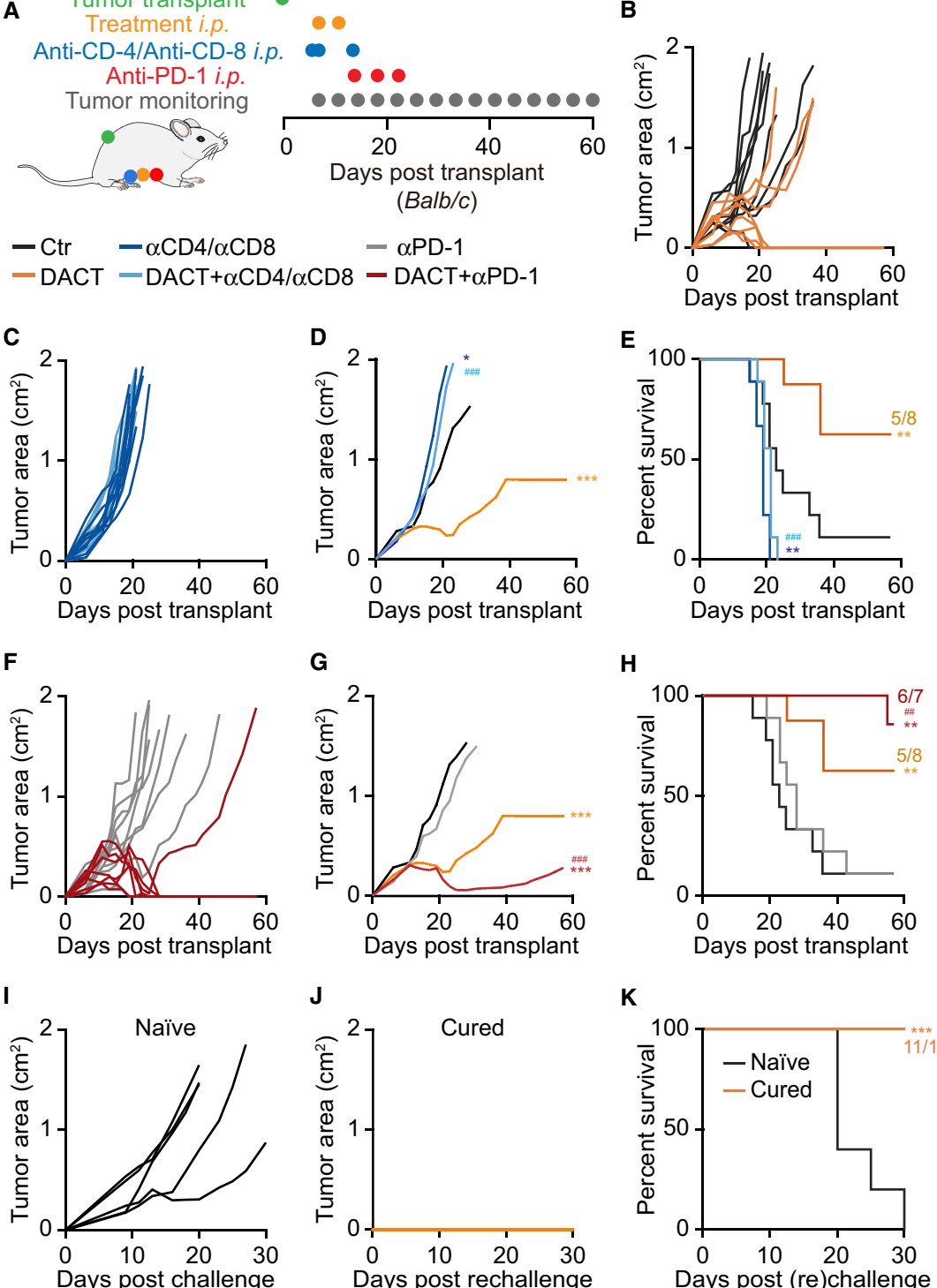

Figure 5.

**Figure 5.  Immune-dependent effect of DACT on WEHI 164 tumors growth and sensitization to immunotherapy.**

A–K   $3 \times 10^5$ mouse fibrosarcoma WEHI 164 cells were injected subcutaneously (*s.c.*) into the flank of immunocompetent syngeneic Balb/c mice with *n* mice per group (*n* = 7 for DACT + anti-PD-1, *n* = 8 for DACT, and *n* = 9 for Ctr, anti-PD-1, anti-CD4/anti-CD8, and DACT + anti-CD4/anti-CD8). When tumors became palpable, the mice were injected intraperitoneally (*i.p.*) with solvent control (Ctr) or with 0.5 mg/kg dactinomycin (DACT). A second injection of chemotherapy was performed 4 days later. Anti-CD4 and anti-CD8 were administered *i.p.* at days −1, 0, and 7 days before/after chemotherapy and anti-PD-1 at days 8, 12, and 16 (A). Tumor size was assessed regularly, and individual tumor growth curves of DACT versus Ctr (B), DACT + anti-CD4/CD8 versus anti-CD-4/anti-CD-8 (C), and DACT + anti-PD-1 versus anti-PD-1 (F) are depicted. Mean tumor area for each group was calculated, and significances were tested using a type II ANOVA test (D, G). Overall survival is depicted, and *P*-values were calculated with a log-rank test (E, H). Stars indicate the *P*-values of each treatment versus Ctr (D, E, G, H), and hashes indicate the *P*-values of the DACT + anti-CD-4/anti-CD-8 versus DACT alone (D, E) and of DACT + anti-PD-1 versus anti-PD-1 alone (G, H) ($*/\#P < 0.05$, $**/\#\#P < 0.01$, $***/\#\#\#\#P < 0.001$). Five naïve mice and the eleven mice that were cured by treatment with DACT alone or in combination with PD-1 blockade were (re)challenged with WEHI 164 cells, and individual tumor growths (I, J), as well as overall survival ($***P < 0.001$, log-rank test) (K), were monitored.

We also addressed the question as to whether ICD inducers must cause reversible or permanent inhibition of anabolic metabolism. For this, we designed a sort of run-on assay using the RUSH (retention using selective hooks) system (Zhao *et al*, 2018) in which an ER-targeted streptavidin protein retains GFP fused to a streptavidin-binding peptide (SBP) in the ER lumen. In the presence of biotin, the GFP signal is lost from the cells because most of the protein is released through the classical Golgi-dependent secretory pathway (Gomes-da-Silva *et al*, 2018; Zhao *et al*, 2018). However, upon washing (to remove biotin) and addition of avidin (to scavenge free biotin), neo-synthesized SBP-GFP is retained by streptavidin in the ER, leading to a progressive increase in fluorescence that directly measures protein synthesis (Fig EV4B). Anthracyclines, oxaliplatin, crizotinib, DACT, and lurbinectedin, which is known to inhibit transcription and has been recently described as an ICD inducer (Tumini *et al*, 2019; Xie *et al*, 2019), largely prevented protein synthesis when continuously present in the system (Fig EV4C–F). After its washout, cells could recover from crizotinib-mediated suppression of protein synthesis. In contrast, the washout of anthracyclines, DACT, oxaliplatin, or lurbinectedin did not lead to the reestablishment of protein synthesis (Fig EV4C–E and G). In sum, ICD-stimulatory anticancer drugs mediate inhibition protein synthesis, though with distinct degrees of reversibility.

**Inhibition of transcription as an ICD hallmark**

In the next step, we addressed the question as to whether inhibition of RNA synthesis would be a general predictor of ICD. For this, we evaluated a homemade library of commonly used antineoplastic agents (Bezu *et al*, 2018) for their capacity to inhibit RNA synthesis using the EU-based assay (Appendix Table S1). We then correlated the level of transcriptional inhibition with the *in silico* ICD prediction score (Fig 8A), all major ICD hallmarks (Fig 8B–E) and their integration into the "ICD score" (Bezu *et al*, 2018) (Fig 8F). Note that vinca alkaloids affected the microscopical assessment of ICD parameters, due to their effect on microtubules, and behaved as outliers, as they failed to inhibit RNA synthesis, yet scored high in predicted ICD-related parameters. After their exclusion, all associations between RNA synthesis inhibition and ICD-related stress/death signals were significant, though with the highest Pearson coefficients for the correlation between, on one side, transcription inhibition and, on the other side, HMGB1 release ($R = 0.76$) or eIF2α phosphorylation ($R = 0.71$). Inhibition of transcription and translation was correlated (Fig 8G), and inhibition of translation was correlated with most ICD hallmarks (Fig EV5A–F, Appendix Table S1). We also went back to the initial *in silico* screen (Fig 1) and compared the five compounds

with the highest ICD prediction score having an $IC_{50} < 1$ μM and that had been introduced into clinical assays (becatecarin, DACT, topotecan, trabectedin, and UCN-01), as well as the six selected compounds with an ICD prediction score close to zero, always with an $IC_{50} < 1$ μM and a clinical characterization (dactolisib, 5-fluoro-deoxycitidine, β-lapachone, mycophenolate mofetil, nonoxynol-9, and RH-1). Consistently, if used at their $IC_{60}$ (Appendix Fig S1) on U2OS cells, the five compounds with a high ICD prediction score were more efficient in suppressing RNA synthesis than the six compounds with a low ICD prediction score (Fig 8H, Appendix Table S2). We then subjected the 50,000 compound library (Shoemaker, 2006) to data mining to identify agents that are annotated as inhibitors of transcription ($n = 31$) or translation ($n = 25$) or, as internal controls, as PARP inhibitors ($n = 4$) or antimetabolites ($n = 45$) (Appendix Table S3). The calculated ICD prediction score was significantly higher than expected for transcription and translation inhibitors but not for other categories of agents such as PARP inhibitors and antimetabolites (Fig 8I–L, Appendix Fig S6A–D). This observation lends further support to the idea that the inhibition of transcription/translation is a major hallmark of ICD.

## Discussion

In the present work, we identified dactinomycin (DACT) as an ICD inducer, using a theoretical prediction that was based on physicochemical characteristics (the "ICD prediction score") and then validated its capacity to induce surrogate hallmarks of ICD in cultured cells and to kill cancer cells in a way that they elicit an antitumor immune response in mice. DACT, like other ICD inducers, can synergize with immunotherapy due to a stimulatory action on the immune system. ICD inducers promote the release or surface exposure of DAMPs, thereby providing adjuvant signals for adaptive anticancer immune responses mediated by T lymphocytes. Subsequent immune checkpoint blockade prevents T-cell exhaustion and improves therapeutic outcome (Pfirschke *et al*, 2016; Serrano-Del Valle *et al*, 2019). The present data could thus lead to a regain of interest for DACT, which is currently limited to some cases of pediatric sarcoma, gestational trophoblastic diseases, and metastatic testicular carcinomas, even though it has shown efficiency in preclinical studies performed in other kinds of cancers (Takusagawa *et al*, 1982; Kam & Thompson, 2010; Cortes *et al*, 2016). Interestingly, one clinical trial currently investigating the effect of DACT in combination with ipilimumab and melphalan for the treatment of melanoma has shown promising results (NCT01323517) (Ariyan *et al*, 2018).

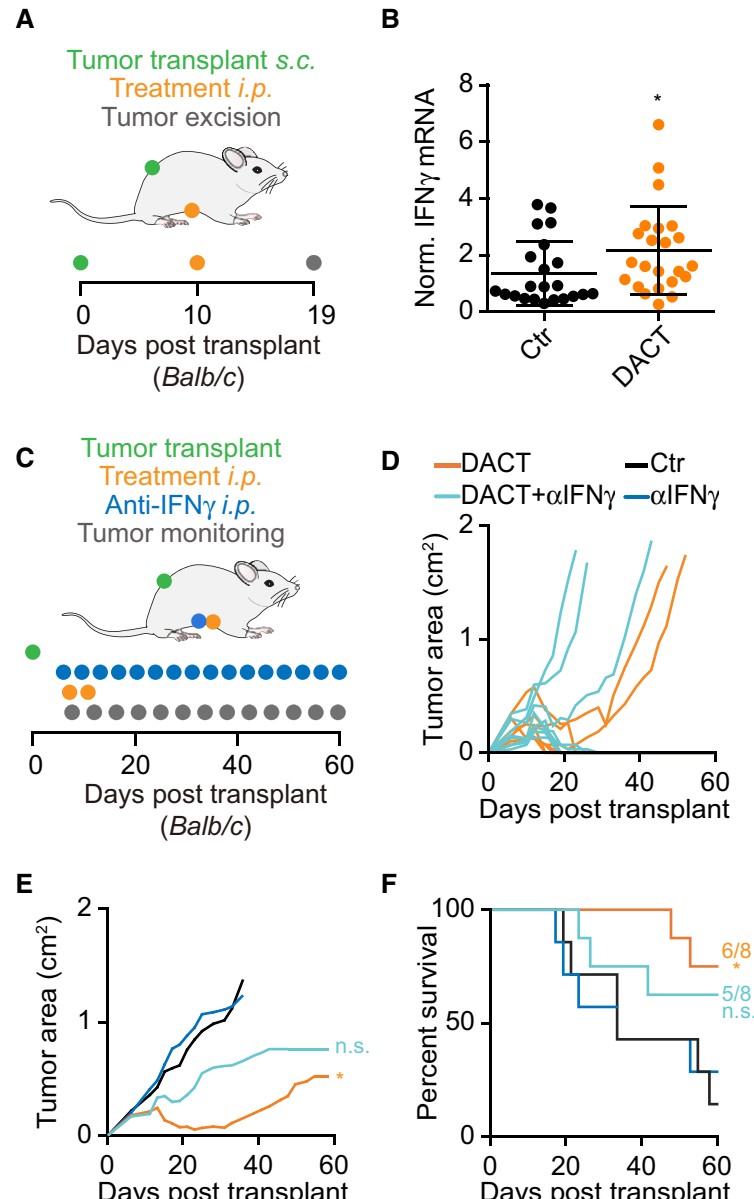

**Figure 6. Role of IFNγ in the anticancer effect of DACT.**

A, B  Mice bearing WEHI 164 sarcomas were treated by systemic (intraperitoneal, *i.p.*) injections of DACT or PBS as a control (Ctr), and tumor was excised 9 days later for the quantitation of mRNA coding for IFNγ (A). Two independent experiments were conducted with a total of 22 mice in the Ctr group and 23 mice in the DACT-treated group, with each data point indicating one tumor. The cycle threshold of the qRT–PCR of IFNγ was normalized to the one of the housekeeping gene peptidylprolyl isomerase A (*Ppia*) in each mouse, and results are shown normalized with respect to controls as a dot plot depicting mean ± SD. The *P*-value (\**P* < 0.05) was calculated by means of Student's *t*-test (B).

C–F  Balb/c mice with palpable WEHI 164 sarcomas (*n* = 7 mice in Ctr and anti-IFNγ groups; *n* = 8 mice in DACT and DACT+anti-IFNγ groups) received two injections of DACT-based chemotherapy (0.5 mg/kg) as well as multiple injections (3 times per week) of neutralizing IFNγ-specific antibody (C). Tumor growth curves are shown for individual mice (D) and as means (E). Statistical difference between DACT-treated tumors and respective controls, which was calculated with a type II ANOVA (\**P* < 0.05), is lost upon IFNγ neutralization (E). Overall survival of mice is also indicated with statistics to respective controls calculated with a log-rank test (\**P* < 0.05) (F).

DACT is well known as an inhibitor of transcription (and actually the standard reagent to block RNA synthesis in wet biology laboratories), and this effect appears to be important for its ICD-inducing activity. Indeed, we found that inhibition of RNA synthesis was a common characteristic of multiple established ICD inducers that differ in their chemical structure, comprising a series of anthracyclines, the tyrosine kinase inhibitor crizotinib, and the platinum salt oxaliplatin. Of note, oxaliplatin (which induces ICD) was more efficient in suppressing transcription than was CDDP, another platinum salt, which is not endowed with strong ICD-inducing capabilities. In

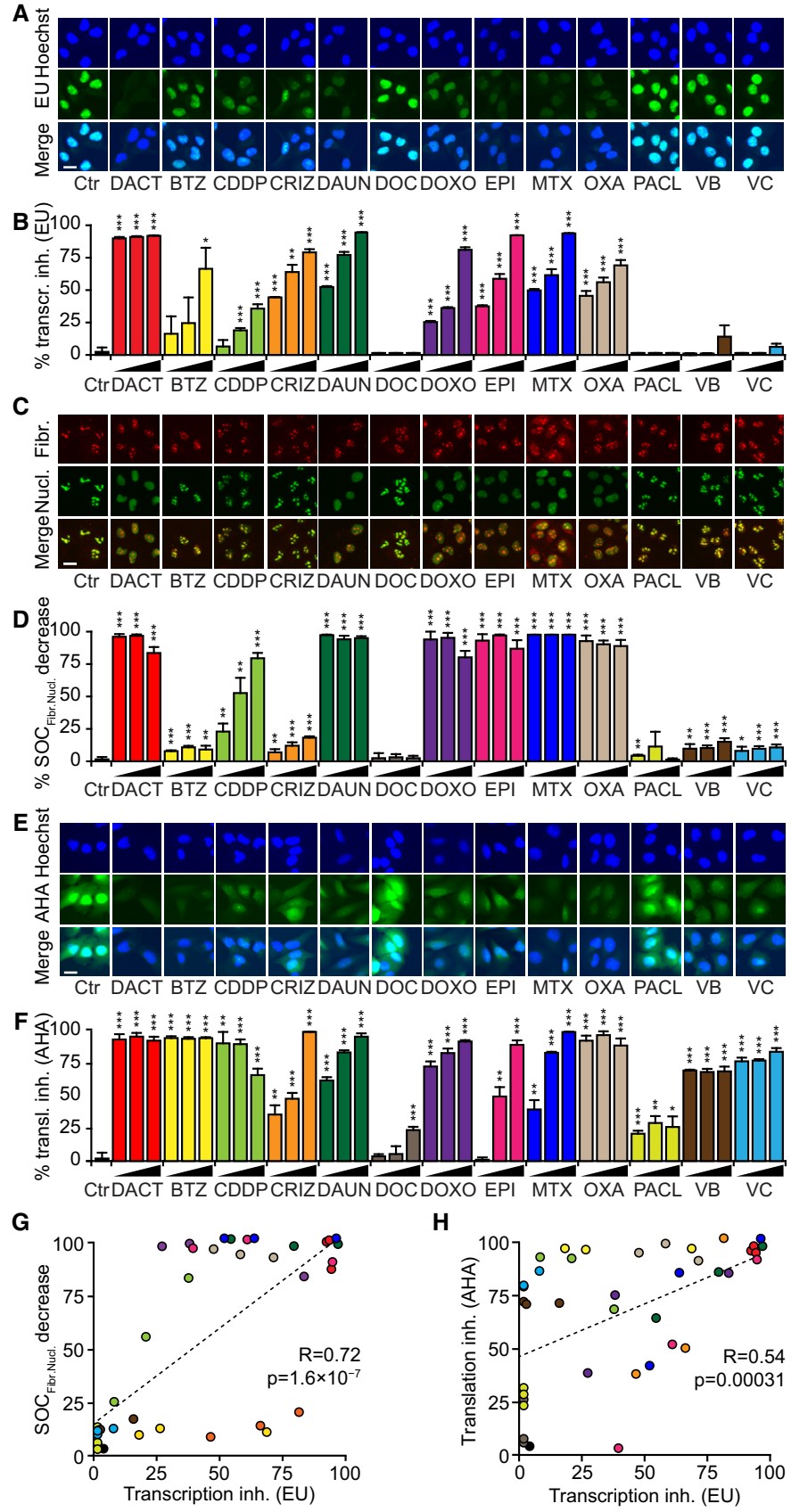

**Figure 7.**

◄ **Figure 7. Inhibition of transcription and translation by ICD inducers.**

A–F Human osteosarcoma U2OS cells were pre-treated with dactinomycin (DACT), bortezomib (BTZ), daunorubicin (DAUN), docetaxel (DOC), doxorubicin (DOXO), epirubicin (EPI), mitoxantrone (MTX), paclitaxel (PACL), vinblastine (VB), and vincristine (VC) at 0.5, 1, and 5 μM; with cisplatin (CDDP) at 75, 150, and 300 μM, with oxaliplatin (OXA) at 250, 500, and 1000 μM, and with crizotinib (CRIZ) at 10, 20, and 40 μM for 1.5 to 2.5 h and followed by an additional hour of treatment in the presence of 100 mM 5-ethynyl uridine (EU). After fixation, cells were permeabilized and EU was stained with an Alexa Fluor-488-coupled azide. Representative images are shown for each treatment (A). The EU intensity in the nucleus of each condition was ranked between the untreated control (Ctr, 0% transcription inhibition) and the control that was not incubated with EU (corresponding to 100% transcription inhibition) (B). Cells were treated for 2.5 h and before fixation and permeabilization. Then, cells were stained with a rabbit anti-fibrillarin antibody followed by a staining with an anti-rabbit Alexa Fluor-647- or Alexa Fluor-546-coupled secondary antibody as well as with a mouse anti-nucleolin antibody followed by a staining with an anti-mouse Alexa Fluor-488-coupled secondary antibody. Then, images were acquired and colocalization between both signals was assessed (C). The surface overlap coefficient (SOC) was calculated and ranked between the untreated control (Ctr) and the positive control (DACT) (D). Cells were pre-treated overnight with the aforementioned compounds in complete medium followed by washout and treatment pursued in methionine-free medium for 30 min. Afterward, the treatments were continued in methionine-free medium supplemented with 25 μM L-azidohomoalanine (AHA) for additional 1.5 h. AHA incorporation was detected after fixation, permeabilization, and blocking by the addition of an Alexa Fluor-488-coupled azide. Then, images were acquired (E) and AHA intensity in the cells was ranked between the untreated control (Ctr, 0% translation inhibition) and the untreated control without AHA (corresponding to 100% translation inhibition) (F).

G, H The correlation between the transcription measured by EU incorporation and measured by fibrillarin and nucleolin colocalization is depicted with Pearson correlation coefficient (R) and P-value (P) (G). The same parameters are shown for the correlation between transcription measured by EU incorporation and translation measured with AHA incorporation (H).

Data information: Representative images of DACT 1 μM, BTZ 1 μM, CDDP 150 μM, CRIZ 20 μM, DAUN 0.5 μM, DOC 1 μM, DOXO 1 μM, EPI 1 μM, MTX 1 μM, OXA 500 μM, PACL 1 μM, VB 1 μM, and VC 1 μM are shown (A, C, E). Scale bars represent 20 μm. One representative experiment among three is shown mean ± SD, and P-values indicating differences to controls were calculated with Student's t-test: *$P < 0.05$, **$P < 0.01$, ***$P < 0.001$ (B, D, F).

addition to DACT, agents that had an elevated ICD prediction score calculated *in silico* (UCN-01, trabectedin, becatecarin, topotecan) turned out to have a significantly higher capacity to suppress RNA synthesis than agents with low ICD prediction scores. These agents widely differ in their chemical structure and mode of action: Topotecan and becatecarin bind to DNA and form a complex with topoisomerases I, trabectedin binds to the minor groove of DNA where it impairs the activity of both transcription factors and polymerases as it promotes alkylation (Tumini *et al*, 2019), and UCN-01 is a cell-permeable staurosporine-derived anticancer agent that inhibits various protein kinases. The importance of the inhibition of transcription in immunogenic cells stress has been further confirmed by the *in silico* analysis of 50,000 agents of the NCI-60 library, revealing a correlation between the "ICD prediction score" and the inhibition of RNA or protein synthesis. Importantly, the reversibility of inhibition apparently has no impact on immunogenicity, but may be linked to the reversibility of the cytotoxic effect of the drugs, as exemplified by the fact that anthracyclines induce a close-to-irreversible

inhibition of protein synthesis, while crizotinib provokes a reversible phenotype (Liu *et al*, 2019).

We previously reported that the pathognomonic hallmark of ICD is the induction of eIF2α phosphorylation, without the activation of the other arms of ER stress, which predicts the immunogenicity of therapeutic interventions (Bezu *et al*, 2018; Giglio *et al*, 2018). EIF2α phosphorylation is induced by ICD-stimulatory chemotherapeutics and correlates with CALR exposure, increased tumor infiltration by activated DC and T lymphocytes, as well as with favorable prognosis (Rae-Grant *et al*, 1991; Panaretakis *et al*, 2009; Fucikova *et al*, 2016a,b; Bezu *et al*, 2018; Giglio *et al*, 2018). DACT, like anthracyclines, oxaliplatin, and crizotinib, induces a split ER stress characterized by the phosphorylation of eIF2α response and consequently CALR exposure *in vitro* and *vivo* (Bezu *et al*, 2018; Liu *et al*, 2019). Conversely, it appears that stalling protein synthesis does not require eIF2α phosphorylation and is rather a direct consequence of transcription inhibition, as cells that are treated with ISRIB or bearing a non-phosphorylable eIF2α mutant reduce their protein

**Figure 8. Validation of the inhibition of transcription as a hallmark of ICD at large scale.** ►

A–G U2OS wild-type cells were treated with a custom-made anticancer library as previously described (Bezu *et al*, 2018) at 3 μM, supplemented with 500 μM oxaliplatin (OXA), 150 μM cisplatin, 50 μM resveratrol, and 50 μM spermidine. For assessing transcription, cells were pre-treated for 1.5 h with the library followed by 1 h with the same drugs in which EU was added. For assessing translation, cells were pre-treated with the library for 12 h followed by 30 min in methionine-free medium, before addition of azidohomoalanine (AHA). Percentage of inhibition was calculated and transformed as z-scores. The correlations between transcription inhibition and ICD prediction score (A), peIF2α expression (B), CALR exposure (C), ATP decrease (D), HMBG1 exodus (E), and biological ICD score (F) previously measured and expressed as z-scores (except for ICD prediction score) (Bezu *et al*, 2018), as well as between transcription and translation inhibitions (G), were calculated with the Pearson method giving the correlation coefficient (R) and corresponding P-values (P). Known immunogenic drugs are indicated with colors: dactinomycin (DACT), mitoxantrone (MTX), doxorubicin (DOXO), daunorubicin (DAUN), OXA, docetaxel (DOC), paclitaxel (PACL), vinblastine (VB), vincristine (VC), and vinorelbine (VR) (A-G).

H The inhibition of transcription was assessed for the negative and positive ICD hits identified with the predictive algorithm (Fig 1). U2OS cells were treated with the agents at concentrations corresponding to their IC60: 1 μM dactinomycin (DACT), 50 μM topotecan, 1 μM becatecarin, 0.5 μM trabectedin, 5 μM UCN-01, 30 μM mycophenolate mofetil, 30 μM nonoxynol-9, 25 μM dactolisib, 2.5 μM β-lapachone, 5 μM 5-fluorodeoxycytidine, and 2 μM RH-1 for 1.5 h followed by 1 h with EU. The percentage of transcription inhibition was calculated, and the coefficient of correlation (R) and associated P-value (P) between the percentage of inhibition and the theoretical ICD score was calculated using the Pearson method.

I–L The 50,000 compounds of the NCI-60 library were annotated for different parameters including transcription and translation inhibition. The predicted ICD score was calculated with a previously described model built on artificial intelligence (Bezu *et al*, 2018). Empirical cumulative distribution is plotted in black for all compounds and in red for the compounds falling into the categories of interest which are transcription inhibitors ($n = 31$) (I), translation inhibitors ($n = 25$) (J), as well as two other random categories used as controls, PARP inhibitors ($n = 4$), and antimetabolites ($n = 49$) (K, L). The P-values calculated with a Kolmogorov–Smirnov test are indicated on each graph.

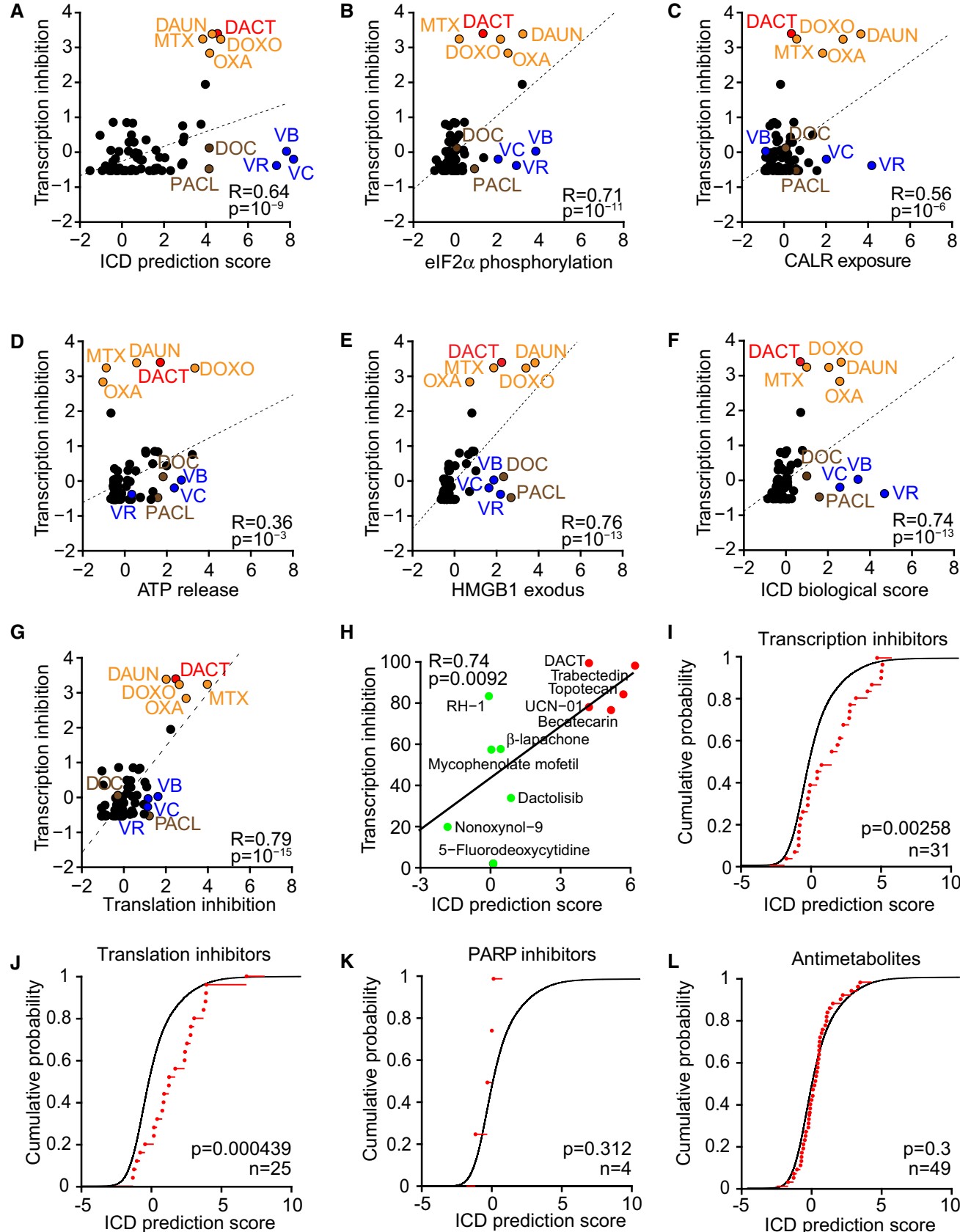

Figure 8.

synthesis in response to DACT and other ICD inducers to the same extent as their wild-type counterparts.

In sum, it appears that inhibition of transcription (and downstream thereof translation) is a common characteristic of ICD inducers. Antineoplastic cytotoxicants exert some degree of "specificity" in the sense that they act more efficiently on cancer cells than on their normal counterparts as well as on tumor-infiltrating leukocytes that participate to immunosurveillance. Sarcomas which are routinely treated with some of the drugs characterized here (such as DACT and trabectedin) are highly sensitive to transcription inhibitors (Jaffe *et al*, 1976; Manara *et al*, 2005; Liebner, 2015; Tumini *et al*, 2019), and it is tempting to speculate that they may be also particularly prone to emit immunogenic signals in response to this kind of anticancer agent. Only a particular vulnerability of cancer cells to transcriptional inhibitors in the context of the "non-oncogene addiction concept" (Luo *et al*, 2009; Nagel *et al*, 2016) may explain why agents that are expected to act on any cell type may elicit a therapeutically relevant stress response without paralyzing vital functions in normal tissues including the immune system.

# Materials and Methods

### Cell lines

Human osteosarcoma U2OS cells were purchased from ATCC. U2OS cells stably expressing HMGB1-GFP together with H2B-RFP and CALR-GFP together with H2B-RFP; GFP-LC3; RFP-LC3; ATF4 reporter; and XBP1-ΔDBD-venus, and U2OS cells co-expressing ss-SBP-GFP and Str-KDEL or HT29 MX1-GFP (in which GFP is under the control of the MX1 promoter) were generated by our group in the past (Shen *et al*, 2012; Zhou *et al*, 2016; Bezu *et al*, 2018; Zhao *et al*, 2018). U2OS cells stably expressing GFP-ATF6 were obtained from Pr. Peter Walter (University of California, San Francisco, USA). U2OS cells stably expressing RFP-LC3 bearing a mutant non-phosphorylable version of eIF2α (eIF2αS51A) were constructed using the CRISPR-Cas9 technology. We designed two complementary gRNAs (Appendix Table S4A) and inserted them into the pX458 vector (containing a tracrRNA and Cas9 fused with 2A-GFP) (Ran *et al*, 2013) following the manufacturer's protocol (New England Biolabs, Ipswich, Massachusetts, USA). We then used this plasmid together with a homology repair template that targets the serine in position 51 of eIF2α for an exchange to alanine (Appendix Fig S6), to transfect RFP-LC3 expressing U2OS cells with Lipofectamine 2000 (Thermo Fisher scientific, Waltham, MA, US) according to the manufacturer's protocol. Two days later, single cells were sorted by flow cytometry. DNA of clones which grew was extracted, amplified by PCR, and analyzed for homozygous knock-in by sequencing (Eurofins Scientific, Luxembourg) (Appendix Table S4B and C). U2OS GFP-LC3 cells having one eIF2α kinase knocked out (EIF2AK1$^{-/-}$, EIF2AK2$^{-/-}$, EIF2AK3$^{-/-}$, and EIF2AK4$^{-/-}$) were constructed using an U6gRNA-Cas9-2A-RFP plasmid containing gRNAs (Sigma-Aldrich, St. Louis, MO, USA) (Appendix Table S4D) following the manufacturer's protocol. In short, U2OS GFP-LC3 cells were transfected and 2 days later, single cells were sorted by flow cytometry. Clones were validated by immunoblot with specific antibodies against human EIF2AK1 (HRI), EIF2AK2 (PKR), EIF2AK3 (PERK), and EIF2AK4 (GCN2) (Appendix Fig S2). Human colon carcinoma HT29 cells and murine lung carcinoma TC-1 cells were purchased from ATCC; mouse fibrosarcoma WEHI 164 cells from Sigma-Aldrich; and mouse fibrosarcoma MCA205 from Merck. All cell lines were regularly tested for the absence of mycoplasma contamination.

### Cell culture

Human osteosarcoma U2OS cells, mouse fibrosarcoma MCA205 cells, human colon adenocarcinoma HT29 cells, and murine lung cancer TC-1 cells were cultured in Dulbecco's modified Eagle's medium (Thermo Fisher Scientific) and mouse fibrosarcoma WEHI 164 cells were cultured in Roswell Park Memorial Institute (RPMI) 1640 medium (ATCC), both supplemented with 10% fetal bovine serum (Gibco by Life Technologies), 1% non-essential amino acids (Thermo Fisher Scientific), and 1% HEPES (Thermo Fisher Scientific) in a humidified incubator with 5% $CO_2$ at 37°C. For U2OS cells co-expressing ss-SBP-GFP and Str-KDEL, 0.25 mg/ml hygromycin and 0.5 mg/ml G418 were added to the culture medium. For U2OS co-expressing HMGB1-GFP and H2B-RFP, 5 μg/ml blasticidin and 0.5 mg/ml G418 were added to the culture medium. For U2OS MX1-GFP, culture medium was supplemented with 2 μg/ml puromycin. Cell culture plastics and consumables were purchased from Corning (NY, USA).

### ICD prediction

The GI$_{50}$ (dose for which 50% of cell growth is inhibited) for a panel of 52,578 compounds (identified via their NSC number) tested on 60 different cell lines was retrieved from the National Cancer Institute website (https://dtp.cancer.gov/databases_tools/bulk_data.htm). From these 52,578 compounds, 49,419 were found to possess a valid PubChem CID (Compound ID number) using the PubChem identifier exchange service (https://pubchem.ncbi.nlm.nih.gov/idexchange/idexchange.cgi). The related structure data file (sdf) for each CID was obtained from Pubchem, and the ICD prediction scores were calculated using the *ICDPred* R package available at https://github.com/kroemerlab/ICDpred. Mitoxantrone was used as a reference to select potential ICD inducers.

### Compounds

A custom-arrayed anticancer library was used (Bezu *et al*, 2018). In addition, bortezomib (5043140001); cisplatin (C2210000); crizotinib (PZ0191); dactinomycin (A1410); daunorubicin (D0125000); docetaxel (01885); doxorubicin (D1515); epirubicin (E9406); flavopiridol (F30055); ISRIB (SML0843); β-lapachone (L2037); mitoxantrone (M6545); mycophenolate mofetil (SML0284); nonoxynol-9 (542334); paclitaxel (T7191); thapsigargin (T9033); tunicamycin (T7765); vinblastine sulfate (V1377); and vincristine sulfate (V0400000) have been bought from Sigma-Aldrich. Dactolisib (BEZ235) (sc-364429) came from Santa Cruz biotechnology (Dallas, TX, USA). Oxaliplatin came from Accord Healthcare (Ahmedabad, India). Topotecan (609699), 7-hydroxystausporine (UCN-01) (72271), becatecarin (101524), 5-fluorodeoxycytidine (B86) (515328), and RH-1 (394347) were kindly provided by the National Cancer Institute (NCI). Lurbinectedin (PM01183) came from PharmaMar (Madrid, Spain).

## Antibodies

Rabbit polyclonal antibodies against CALR (ab2907), rabbit monoclonal phosphoneoepitope-specific antibody against phospho-eIF2α (Ser51) (ab32157, clone E90), rabbit polyclonal antibody against fibrillarin (ab5821), mouse monoclonal antibody against nucleolin (ab13541, clone 4E2), and mouse monoclonal antibody against β-actin (ab49900, clone AC-15) were purchased from Abcam (Cambridge, UK). Rabbit polyclonal antibody against HRI (sc-30143) and mouse monoclonal antibody against PKR (sc-6282, clone B-10) were purchased from Santa Cruz biotechnology. Rabbit monoclonal antibody against PERK (#3192, clone C33E10) and rabbit polyclonal antibody against GCN2 (#3302) came from Cell Signaling Technology (Danvers, MA, USA). Depleting or neutralizing antibodies for *in vivo* purpose: anti-PD-1 (BE0273, clone 29F.1A12), anti-CD4 (BE0003-1, clone GK1.5), anti-CD8a (BE0061, clone 2.43), rat IgG2a anti-trinitrophenol isotype control (BE0089, clone LTF-2), anti-IFNγ (BE0054, clone R4-6A2), and rat IgG1 anti-horseradish peroxidase isotype control (BE0088, clone HRPN), were purchased from BioXcell (West Lebanon, NH, USA). Anti-rabbit and anti-mouse Alexa Fluor-488, Alexa Fluor-568, and Alexa Fluor-647 secondary antibodies came from Thermo Fisher Scientific. Conjugated antibodies for flow cytometric analysis of immune receptors were purchased from BD Pharmingen (Franklin Lakes, NJ, USA), BioLegend (San Diego, CA, USA), or Miltenyi Biotec (Bergisch Gladbach, Germany).

## Fluorescence microscopy, image acquisition and analysis

One day before treatment, 2,500 U2OS cells either wild-type or stably expressing ATF6-GFP, ATF4-GFP, or XBP1-ΔDBD-venus were seeded in 384-well μClear imaging plates (Greiner Bio-One) and let adhere. The next day, cells were treated for 6, 12, and 24 h to assess ATP decrease, 6 h to look at eIF2α phosphorylation (peIF2α) and ATF6, or 12 h to assess ATF4 and spliced XBP1 (XBP1s) levels. Next, cells were fixed with 3.7% formaldehyde (F8775, Sigma-Aldrich) supplemented with 1 μg/ml Hoechst 33342 (H3570, Thermo Fisher Scientific) for 1 h at room temperature. For ATF6, ATF4, and XBP1s, the fixative was exchanged to PBS and the plates were analyzed by automated microscopy. EIF2α phosphorylation was assessed by immunostaining: To this aim, cells were treated and fixed as described above in the presence of Hoechst 33342. Then unspecific antibody interaction was blocked by 2% BSA for 1 h at room temperature and followed by an incubation with 1:500 antibody specific for phospho-eIF2α (Ser51) overnight at 4 °C. After several washing steps with PBS, cells were stained with 1:1,000 Alexa Fluor-568-coupled secondary antibody for 2 h at room temperature and washed with PBS before acquisition. For the detection of ATP enriched vesicles, the cells were labeled after treatment with the fluorescent dye quinacrine as described before (Martins *et al*, 2011). Briefly, cells were incubated with 5 μM quinacrine and 1 μg/ml Hoechst 33342 in Krebs–Ringer solution (125 mM NaCl, 5 mM KCl, 1 mM $MgSO_4$, 0,7 mM $KH_2PO_4$, 2 mM $CaCl_2$, 6 mM glucose, and 25 mM HEPES, pH 7.4) for 30 min at 37°C. Thereafter, cells were rinsed with Krebs–Ringer and viable cells were microscopically examined. For automated fluorescence microscopy, a robot-assisted Molecular Devices IXM XL BioImager and a Molecular Devices IXM-C (Molecular Devices, Sunnyvale, CA, USA) equipped with either a SpectraX or an Aura II light source (Lumencor,

Beaverton, OR, USA), adequate excitation and emission filters (Semrock, Rochester, NY, USA) and a 16-bit monochromes sCMOS PCO.edge 5.5 camera (PCO Kelheim, Germany) or an Andor Zyla camera (Belfast, Northern Ireland) and a 20× PlanAPO objective (Nikon, Tokyo, Japan) were used to acquire a minimum of four view fields per well, followed by automated image processing with the custom module editor within the MetaXpress software (Molecular Devices). Image segmentation was performed using the MetaXpress software (Molecular Devices). The primary region of interest (ROI) was defined by a polygon mask around the nucleus allowing for the enumeration of cells, the detection of morphological alterations of the nucleus and nuclear fluorescence intensity. Secondary cytoplasmic ROIs were used for the quantification of quinacrine, peIF2α, XBP1s, and ATF6. After exclusion of cellular debris and dead cells from the dataset, parameters of interest were normalized, statistically evaluated, and graphically depicted with R software. Using R, images were extracted and pixel intensities scaled to be visible (in the same extent for all images of a given experiment). Scale bars represent 20 μm, except for microphotographs of tissue (Fig 3K and M) where it represents 10 μm.

## CALR translocation and HMGB1 release by video microscopy

One day before treatment, 2,500 U2OS cells stably co-expressing either CALR-GFP or HMGB1-GFP with H2B-RFP per well were seeded in 384-well μClear imaging plates (Greiner Bio-One) and let adhere. The next day, cells were treated and CALR-GFP and HMGB1-GFP cells were observed by live-cell microscopy as described before with a frequency of image acquisition at one image per hour for 12 and 24 h, respectively. The images were segmented and analyzed with R using the *EBImage* and *flowcatchR* packages from the Bioconductor repository (https://www.bioconductor.org). H2B-RFP was used to segment nuclei, and the obtained mask was either used to measure GFP intensity in the nuclear compartment (HMGB1-GFP) or as a seed to segment the cytoplasmic compartment (CALR-GFP). Then, a top-hat filter was applied and the area of CALR-GFP$^{high}$ regions was measured. HMGB1-GFP nuclear fluorescence intensity of single cells tracked over time was normalized to its value at first time point.

## Determination of $IC_{60}$

U2OS wild-type cells were seeded at 8,000 cells per well in 96-well μClear imaging plates (Greiner Bio-One) and let adhere for 24 h before treatment. Cells were treated with a large range of concentrations for 24 h and then stained by the addition of propidium iodide (P4864, Sigma-Aldrich) at a final concentration of 1 μg/ml and Hoechst 33342 at 2 μg/ml for 30 min. Plates were centrifuged in order to bring detached cells to the focal plane and then, images were acquired by automated microscopy using adequate filter sets as described above. The images were segmented with R by means of the *EBImage* package. Nuclei were segmented based on Hoechst 33342 signal; then, nuclear area and fluorescence intensities (in DAPI and Cy3) were measured. The assessed parameters were used to cluster cells as healthy (normal-sized, Hoechst$^{low}$, PI$^−$), pyknotic (condensed, Hoechst$^{high}$, PI$^−$), or dead (PI$^+$). The number of healthy cells was then used to establish dose–response models, by fitting the data points with a 4-parameter log-logistic model. The

model was then used to calculate the $IC_{60}$ (concentration for which 40% of cell population is healthy) for each drug.

## MX1 pathway activation

This technique was previously developed in our laboratory (Zhou et al, 2016). U2OS or MCA205 wild-type cells were seeded at 8,000 cells per well in 96-well μClear imaging plates (Greiner Bio-One) and let adhere for 24 h. Next, cells were treated for 6 h and medium was changed for the following 24 h. Afterward, the supernatant of each condition was transferred on HT29 MX1-GFP plated at 4,000 cells per well in 96-well μClear imagine plates 2 days before. As an additional control, HT29 MX1-GFP was treated with IFNα1 (752802, BioLegend). Forty-eight hours later, the plates were fixed with 3.7% formaldehyde supplemented with 1 μg/ml Hoechst 33342 for 1 h at room temperature. The fixative was exchanged to PBS, and the plates were analyzed by automated microscopy. The amount of GFP intensity in the whole cell was measured, and the number of positive cells was calculated based on a threshold set between the distribution of the GFP intensity in untreated cells and the one in IFNα1-treated cells.

## Protein immunoblot

Protein was extracted with RIPA buffer (#89900; Thermo Fisher Scientific) in the presence of phosphatase and protease inhibitors (#88669; Thermo Fisher Scientific) followed by sonication. Then, protein content was measured by Bio-Rad laboratory DC Protein Assay (#500-0113, #500-0114 and #500-0115 Thermo Fisher Scientific) following the manufacturer's protocol. 20 μg of protein was dissolved in Laemmli buffer (Thermo Fisher Scientific), denatured at 100°C, and separated by polyacrylamide gel electrophoresis (PAGE) using 4–12% Bis-Tris pre-casted gels (Thermo Fisher Scientific) in MOPS buffer (Thermo Fisher Scientific). Afterward, proteins were transferred to EtOH-activated PVDF membranes (Merck Millipore IPVH00010) in transfer buffer (25 mM Tris; 190 mM glycine; 20% methanol in $H_2O$) at 200 mA and 120 V for 1.5 h. Membranes were washed in Tris-buffered saline with Tween20 buffer (TBST; 20 mM Tris, pH 7.5 150 mM NaCl 0.1% Tween 20 in $H_2O$) and then blocked with 5% BSA in TBST for 1 h. Membranes were exposed to primary 1:1,000 antibody diluted in 5% BSA in TBST overnight at 4 °C. Next, membranes were washed three times with TBST and then were incubated with 1:5,000 appropriate horseradish peroxidase-coupled secondary antibody (Southern Biotech, Birmingham, AL, USA) for 1 h at room temperature. Proteins were revealed with ECL (GE Healthcare, Chicago, Il, USA). Beta-actin (at 1:10,000) was used to verify equal loading.

## Assessment of CALR exposure by flow cytometry

U2OS wild-type cells were seeded at 8,000 cells per well in 96-well plates. The cells were treated for 6 h, and the drug was washed out. Twenty-four hours later, the cells were collected and ecto-CRT was detected by immunofluorescence staining. Cells were incubated for 30 min at 4 °C with 1:100 primary antibody specific for CALR and then washed and further incubated with 1:500 secondary Alexa Fluor-488-coupled anti-rabbit antibody for 30 min at 4°C. Finally, 4′, 6-diamidino-2 phenylindole dihydrochloride (DAPI, # D1306,

Thermo Fisher Scientific) was added before flow cytometric analysis. Samples were analyzed using a CyAn ADP cytofluorometer (Beckman Coulter, Brea, CA, USA) coupled to a HyperCyt loader (IntelliCyt, Albuquerque, NM, USA). Alternatively, after incubation with primary antibody, cells were stained with the LIVE/DEAD Fixable dead cell stain (Thermo Fisher Scientific) and fixed with 3.7% formaldehyde. After staining with the secondary antibody, they were acquired with a MacsQuant cytometer (Miltenyi Biotec). Using FlowJo v10 software (TreeStar, Inc.), the percentage of $CALR^+$ cells among $DAPI^-$ cells is quantified.

## Assessment of extracellular ATP

The ENLITEN ATP Bioluminescence Detection Kit (FF2000; Promega, Madison, MI, USA) was used for measurement of ATP in cell culture supernatants. Briefly, 8,000 U2OS or MCA205 wild-type cells per well were seeded in 96-well plates. The following day, the cells were treated and 24 h later, the supernatant was collected and centrifuged. The supernatant was transferred to a white bottom plate, and the enzyme and substrate from the kit were added. ATP-dependent substrate conversion was measured by assessing luminescence at 560 nm in a SpectraMax I3 multi-mode plate reader (Molecular Devices).

## Assessment of extracellular HMGB1

The ELISA kit (ST51011; IBL International GmbH, Hamburg, Germany) was used for measuring HMGB1 released in the supernatant. Briefly, 8,000 U2OS or MCA205 wild-type cells per well were seeded in 96-well plates. The following day, the cells were treated and 24 h later, the supernatant was collected and centrifuged. ELISA was performed as described by the provider, and absorbance at 450 nm was assessed with a SpectraMax I3 multi-mode plate reader (Molecular Devices).

## Assessment of BMDC-mediated phagocytosis

To measure phagocytosis, we used standard protocol as previously described (Cerrato et al, 2020). Bone marrow-derived dendritic cells (BMDCs) were generated from femurs and tibias taken from C57Bl/6 mice. Bone marrow was collected by flushing the bones with PBS, and clusters were dissolved by pipetting. Red blood cells were lysed with red cell lysis buffer (0.01 M Tris, 0.83% $NH_4Cl$ in Milli-Q water). After washing and filtration through a 70-μm cell strainer, cells were seeded into 6-well plates at a density of $1.5 \times 10^6$ viable cells in 2 ml of BMDC culture medium (RPMI 1640 medium supplemented with 10% FBS, 100 U/ml penicillin, 1 M HEPES, 1× MEM Non-Essential Amino Acids Solutions, and 50 μM β-mercaptoethanol) supplemented with 20 ng/ml recombinant mouse GM-CSF and 5 ng/ml IL-4. At day 3, additional 1 ml of complete medium was added to each well. At day 6, half of the supernatant was removed, centrifuged, and added back to the original culture with the addition of 5 ng/ml IL-4 and 20 ng/ml GM-CSF. Non-adherent and loosely adherent BMDCs were harvested on day 7, counted, and $8 \times 10^5$ mouse fibrosarcoma MCA205 cells were seeded in standard 25 $cm^2$ polystyrene flasks for cell culture. Twenty-four hours later, they were labeled with 0.5 μM CellTracker Orange CMTMR (5-(and-6)-(((4-chloromethyl)benzoyl)amino)

tetramethylrhodamine) dye diluted in serum-free medium according to the manufacturer's protocol (Thermo Fisher Scientific). MCA205 cells were further treated for 24 h before co-culture with BMDCs in 6-well plates at 37 or 4°C at a 1:4 ratio (BMDC:MCA205). After 4 h, cells were detached with a cell lifter, and BMDCs were stained with conjugated 1:200 anti-CD11c FITC antibody (clone N418, BioLegend) diluted in 1% BSA in PBS and incubated at 4°C in the dark for 30 min. Cells were washed and fixed in 3.7% formaldehyde in PBS. Samples were run through a BD LSRFortessa flow cytometer, and data were acquired using BD FACSDiva software (BD Biosciences). Phagocytosis efficiency was assessed by measuring the ratio of $CMTMR^+ CD11c^+$ cells among total amount of $CD11c^+$ BMDCs, using FlowJo software.

## Assessment of DCs activation markers

BMDCs were collected and differentiated as described above. MCA205 were treated for 24 h and co-cultured with BMDCs (100,000 dying MCA205 cells and 100,000 BMDCs per well in round-bottom 96-well plate) for 24 h at 37°C. Subsequently, cells were incubated with Fixable Viability Dye eFluor 780 (Invitrogen) and anti-mouse CD16/CD32 (clone 93, BioLegend) to block Fc receptors for 15 min, at 4 °C, and then stained with 1:100 anti-CD11c PerCP Cy5.5 (clone N4/8, BioLegend) or anti-CD11c eF450 (clone HM450, eBioscience) as well as with 1:400 anti-MHCII FITC (clone M5/114.15.2, BioLegend), and anti-CD86 APC (clone GL-1, BioLegend) for 30 min at 4 °C before fixation with 3.7% formaldehyde. Samples were run through a BD LSRFortessa flow cytometer, and data were acquired using BD FACSDiva software (BD Biosciences) and analyzed with FlowJo software.

## *In vivo* experimentation

Six- to eight-week-old female wild-type C57Bl/6, Balb/c, and *nu/nu* mice were purchased from Envigo (Huntingdon, UK) and were housed in the animal facility at the Gustave Roussy Cancer Center in a pathogen-free, temperature-controlled environment with 12-h day and night cycles and received water and food *ad libitum*. Animal experiments were conducted in compliance with the EU Directive 63/2010 and with protocols 20180712102764 51_n2018_051_16095, 201903131451670_n2019_017_19749, or 2019072311495586_n2019_050_21586 and were approved by the Ethical Committee of the Gustave Roussy Cancer Center (CEEA IRCIV/IGR no. 26, registered at the French Ministry of Research). For mouse experiments, "BiostaTGV" software was used to calculate the number of animals in each group needed to reach statistical significance, based on expected results. Mice were randomized in different groups based on tumor size just before the injection of the first treatment. Tumor length and width were assessed with a standard caliper 2 to 3 times a week, and tumor areas were calculated using the formula of an ellipse: L × l × 3.14 / 4. No blinding was done. The mice were sacrificed when tumors reached 1.8 $cm^2$ or depicted any signs of discomfort following the EU Directive 63/2010 and our Ethical Committee advice (or before, if required by the setting of the experiment). Animals that had to be sacrificed during the experiment due to an ethical endpoint other than tumor size were excluded from the analysis. Tumor growth and overall survival were analyzed with the help of the TumGrowth software

package (Enot *et al*, 2018), freely available at https://github.com/kroemerlab.

## *In vivo* xenograft experiments

*Nu/nu* mice were administered $5 \times 10^6$ U2OS CALR-RFP cells subcutaneously (*s.c.*) into the flank of. After 6–10 weeks, when tumors became palpable, 200 μl of the agents diluted in PBS were administered intratumorally. Tumors were resected after 6 or 24 h of treatment, immediately fixed in 3.7% formaldehyde overnight, and then incubated in sucrose gradients. Tumors were sectioned into 5-μM slices using a cryostat microtome. Two slices per tumor were stained with an antibody specific for phospho-eIF2α (Ser51) used at 1:500 in 2% BSA overnight at 4°C after blocking with 2% BSA. Then, slides were washed and incubated with 1:1,000 Alexa Fluor-647-coupled secondary antibody for 2 h at room temperature and counterstained with Hoechst 33342. Images were acquired with an IXM-C confocal microscope (Molecular Devices) as described above with a 10-fold magnification. After exclusion of out-of-focused images, segmentation was conducted with R. Nuclei were identified based on Hoechst 33342 staining, and cytoplasmic regions were segmented based on CALR-RFP signal. The intensity of the Cy5 signal in the cytoplasmic region was measured to assess eIF2α phosphorylation. For CALR translocation, the coefficient of variation of the RFP signal in the cytoplasmic region was calculated, which represents the standard deviation divided by the mean of the intensity of the signal.

## Anticancer vaccination

For anticancer vaccination studies, we used standard protocol as described before (Casares *et al*, 2005; Humeau *et al*, 2019). In brief, MCA205 cells were treated for 24 h to reach 50–70% mortality, and then, $1 \times 10^6$ cells were resuspended in 200 μl PBS and injected *s.c.* into the left flank of immunocompetent C57Bl/6 animals. Two weeks later, $1 \times 10^5$ living MCA205 cells were injected *s.c.* in the other flank and tumor growth was monitored for the forthcoming weeks.

## *In vivo* tumor treatment

Established tumors were assessed for their response to DACT-based chemotherapy. To this aim, MCA205 fibrosarcoma cancers were established *s.c.* in C57Bl/6 mice and in *nu/nu* mice by injection of $3 \times 10^5$ MCA205 cells. When tumors became palpable, 200 μl of the chemotherapeutics (diluted in 10% PEG 400, 10% Tween 70, 4% DMSO, and 7% NaCl) or the diluent alone were injected intraperitoneal (*i.p.*) and tumor growth was monitored for the forthcoming weeks.

Alternatively, WEHI 164 fibrosarcoma tumors were established *s.c.* in Balb/c mice by injection of $3 \times 10^5$ MCA205 cells. When tumors became palpable, 200 μl of the chemotherapeutics or the diluent alone were injected *i.p.* Four days later, the treatment was repeated. To study the importance of the adaptive immune system, 100 μg CD4 and CD8a specific depleting antibodies or corresponding isotype diluted in PBS were injected at days -1, 0, and 7 before and after the first injection of chemotherapy. To evaluate the role of IFNγ, 100 μg IFNγ blocking antibody or corresponding isotype was

injected three times per week, starting on the day before the first chemotherapy, until experimental endpoints were reached.

## Combination with immunotherapy

In order to boost the effect of the chemotherapeutics, the same treatments were combined sequential with immune checkpoint blockade. MCA205 fibrosarcoma cell cancers were established *s.c.* in C57Bl/6 mice by injection of $1 \times 10^5$ MCA205 cells. When tumors became palpable, the chemotherapeutics were injected *i.p*. At days 6, 10, and 14 after chemotherapy, 200 μg/mouse of anti-PD-1 or corresponding isotype, prepared in 200 μl PBS, was injected *i.p*. Alternatively, when used to treat WEHI 164 tumors (established *s.c.* in Balb/c by injection of $3 \times 10^5$ cells), 200 μg/mouse of anti-PD-1 was injected at days 8, 12, and 16 after the first chemotherapy. Tumor growth was monitored for the forthcoming weeks.

Surviving C57Bl/6 animals that were tumor-free after treatment were analyzed for the generation of immunological memory by *s.c.* rechallenge with $1 \times 10^5$ non-isogenic TC-1 cells in the flank initially injected and $1 \times 10^5$ isogenic MCA205 cells injected in the contralateral flank. Naïve C57Bl/6 mice were used as control. Surviving Balb/c mice were rechallenged in the contralateral flank with $3 \times 10^5$ isogenic WEHI 164 cells and naïve Balb/c mice were used as controls.

## Study of immune receptors in the tumor infiltrate

MCA205 fibrosarcoma cancers were established subcutaneously (*s.c.*) in C57Bl/6 mice by injection of $2 \times 10^5$ MCA205 cells. When tumors became palpable, 200 μl of the chemotherapeutics were injected *i.p* with $n = 10$ mice per group. Nine days later, tumors were harvested, weighed, and then processed before phenotyping immune cells as described before (Levesque *et al*, 2019). In brief, tumors were dissociated mechanically with scissors and then enzymatically with the tumor dissociation kit (Miltenyi) and the gentle MACS Octo Dissociator (Miltenyi) following the manufacturer's protocol. Tumor cell homogenates were filtered through 70-□m SmartStrainers (Miltenyi) and washed twice with PBS. Tumor cell homogenates, corresponding to 50 mg of the initial tumor sample, were stained with LIVE/DEAD Fixable Yellow dead cell stain (Thermo Fisher Scientific) and then with anti-mouse CD16/CD32 (clone 2.4G2, Mouse BD Fc Block, BD Pharmingen) to block Fc receptors. To determine the production of cytokines by T lymphocytes, sample cells were stimulated for 5 h in serum-free CTL-Test PLUS Medium (ImmunoSpot) containing 20 ng/ml phorbol myristate acetate (PMA, Calbiochem), 1 μg/ml ionomycin (Sigma-Aldrich), and BD Brefeldin A (GolgiPlug, dilution 1:1,000, BD Biosciences). Immune cell staining was performed with the set of fluorochrome-conjugated antibodies related to each panel with first, a staining of surface receptors and second, after incubation in eBioscience FoxP3/Transcription Factor Staining Buffer (Thermo Fisher Scientific) and permeabilization, a staining of intracellular receptors and cytokines:

1  (Panel 1) "*T-cell panel*": anti-CD3g,d,e APC (clone 17A2, BioLegend, dilution 1:400), anti-CD8a PE (clone 53-6.7, BD Pharmingen, dilution 1:400), and anti-CD4 PerCP-Cy5.5 (clone

RM4-5, Thermo Fisher Scientific, dilution 1:800) completed by an intranuclear staining with anti-FoxP3 FITC (clone FJK-16s, Thermo Fisher Scientific, dilution 1:50).

2  (Panel 2) "*NK and cytokines panel*": anti-CD45 BUV661 (clone 30F11, BD Pharmingen, dilution 1:200), anti-CD3g,d,e BV421 (clone 53-8.7, BD Pharmingen, dilution 1:200), anti-CD4 BUV496 (clone GK1.5, BD Pharmingen), anti-CD8a PE (clone 53-6.7, BD Pharmingen), anti-NK1.1 BV605 (clone PK136, BD Pharmingen, dilution 1:100), and anti-TCRγ( BV711 (clone GL3, BD Pharmingen, dilution 1:100) completed by an intracellular staining with anti-IL17a APC-Cy7 (clone TC11-18H10, BD Pharmingen, dilution 1:100) to assess the Th17 response, anti-IFNγ APC (clone XMG1.2, BioLegend) for measuring the Th1/Tc1 response, and anti-IL4 PerCP-Cy5.5 (clone 11B12, BD Pharmingen, dilution 1:100) characterizing a Th2 response.

Stained samples of panel 1 were run through a BD LSR II flow cytometer while for the samples of the panel 2, a BD LSRFortessa flow cytometer was used. All samples were acquired using BD FACSDiva software (BD biosciences) and analyzed using FlowJo software.

## Quantitation of IFNγ mRNA in the tumor infiltrate

WEHI 164 fibrosarcoma cancers were established *s.c.* in Balb/c mice by injection of $3 \times 10^5$ MCA205 cells. Ten days later, 200 μl of the chemotherapeutics were injected *i.p*. Nine days following treatment, tumors were harvested in RNA*later* solution. A small piece of tumor (< 30 mg) was placed in hard tissue homogenizing CK28-R tubes (Bertin, France) and grinded with a Precellys 24 tissue homogenizer (Bertin). RNA was extracted with a RNeasy Plus Mini Kit (Qiagen, Venlo, Netherlands) following the manufacturer's protocol, and 2,500 ng of RNA per condition was subjected to reverse transcription with SuperScript IV VILO Master Mix (Thermo Fisher Scientific). A quantitative PCR of each sample was run in duplicate to measure the amount of mRNAs coding for IFNγ (with probe Mm00439619m1, Thermo Fisher Scientific) and peptidylprolyl isomerase A (*Ppia*) (with probe Mm02342430, Thermo Fisher Scientific) using TaqMan Fast Advanced Master Mix (Thermo Fisher Scientific). qRT–PCR data were normalized to the expression level of the housekeeping gene *Ppia* and then to the means of the level of corresponding mRNA in control mice.

## Evaluation of transcription by EU incorporation

Transcription was analyzed by measuring the incorporation of Click-iT chemistry-detectable 5-ethynyl uridine (EU) (C10327; Thermo Fisher Scientific) following the manufacturer's advice. In short, 2,500 cells per well were cultured in 384-well μClear imaging plates. The next day, cells were pre-treated for 1.5 to 2.5 h and washed and treatment was pursued in the presence of 1 mM 5-ethynyl uridine (EU) for 1 h. Following, the cells were fixed with 3.7% formaldehyde supplemented with 1 μg/ml Hoechst 33342 for 1 h and permeabilized with 0.1% Triton X-100 for 10 min. Alexa Fluor-488-coupled azide was then added for 2 h. The intensity of the GFP signal (EU) in the nucleus was measured by microscopy, and the inhibition of transcription was calculated by ranging the GFP intensity in each condition between its value in untreated cells

### The paper explained

#### Problem

Chemotherapy still constitutes the standard treatment for most cancers. Yet, some chemotherapeutics are able to trigger stress signals in cancer cells, which activate an antitumor immune response and thereby confer long-term protection. We first investigated the immunogenic potential of chemotherapeutics which are already tested in clinics. Then, we further elucidated the mechanisms underlying this effect.

#### Results

A machine learning approach was used for the prediction of novel inducers of immunogenic cell death (ICD) within a library of 50,000 compounds. This approach led to the identification of dactinomycin (DACT) that induces ICD *in vitro* and mediates anticancer immunity *in vivo*. DACT is commonly used as an inhibitor of DNA to RNA transcription. An analysis of established and predicted ICD inducers revealed the inhibition of RNA transcription (and secondarily protein translation) as an initial event for ICD induction.

#### Impact

These findings may improve the application of dactinomycin in clinics and offer new combination strategies for the treatment of childhood sarcoma. In addition, the discovery of transcription as a characteristic of ICD may facilitate the development of immunotherapies.

(0% inhibition) and its value in cells that have not been incubated with EU (corresponding to 100% inhibition).

### Evaluation of transcription by colocalization of fibrillarin and nucleolin

2,500 cells per well were cultured in 384-well µClear imaging plates. The next day, cells were treated for 2.5 h. Following, cells were fixed with 3.7% formaldehyde supplemented with 1 µg/ml Hoechst 33342 for 1 h, permeabilized with 0.1% Triton X-100 for 10 min, and blocked with 2% BSA for 1 h. Cells were further incubated with 1:2,000 rabbit antibody specific for fibrillarin and 1:4,000 mouse antibody specific for nucleolin overnight at 4°C. After several PBS washing steps, 1:1,000 anti-mouse Alexa Fluor-488- and anti-rabbit Alexa Fluor-647 (or Alexa Fluor-546 in mitoxantrone-treated cells to avoid interference of with drugs auto-fluorescence)-coupled antibodies were added. Following several PBS washing steps, the DAPI, GFP, and Cy5 (or Cy3) signals were acquired with a confocal microscope IXM-C (Molecular Devices). The obtained images were analyzed using ColocalizR software (Sauvat *et al*, 2019) taking into consideration the SOC (Surface Overlap Coefficient) between the GFP and Cy5 (or Cy3) signals. Data were then ranged between 0% inhibition (Ctr) and 100% inhibition (corresponding to the well with lowest SOC in the dataset).

### Translation study by AHA incorporation

Translation was measured by assessing the incorporation of L-azidohomoalanine (AHA) (C10289; Thermo Fisher Scientific), a labeled form of methionine by Click-iT chemistry following the manufacturer's advice. In short, 2,000 cells per well were cultured in 384-well µClear imaging plates. The next day, cells were treated for 12 h. After several PBS washing steps, the cells were incubated 30 min in the presence of methionine-free medium. They were further treated 1.5 h in methionine-free medium in the presence of 25 µM AHA. Afterward, the cells were fixed with 3.7% formaldehyde supplemented with 1 µg/ml Hoechst 33342 for 1 h, permeabilized with 0.1% Triton X-100 for 10 min, and blocked with 2% BSA for 1 h. Then, Alexa Fluor-488-coupled azide was added for 2 h and the GFP intensity (AHA) was measured by microscopy. Translation inhibition was calculated by ranging the GFP intensity in each condition between its value in untreated cells (0% inhibition) and its value in cells that have not been incubated with AHA (corresponding to 100% inhibition).

### Reversibility of proteins synthesis assessed by Rush

U2OS cells co-expressing ss-SBP-GFP and Str-KDEL were seeded at 2,500 cell per well in 384-well µClear imaging plates. The following day, cells were stained with 0.5 µM CellTracker Orange (CMTMR) diluted in serum-free medium according to the manufacturer's protocol (Thermo Fisher Scientific) and then pre-treated with 40 µM biotin for 4 h and with the compounds to test for 2.5 h. After washout, cells were treated with 1 µM avidin to assess reversibility of the inhibition of transcription (discontinuous treatments). As a positive control, treatments were pursued in the presence of avidin (continuous treatments). Images were then acquired every hour for 24 h. ROIs were defined based on CellTracker staining, and GFP intensity was quantified. Values were normalized to the control at each time point, and the percentage of inhibition and of reversibility were calculated. For this first parameter, the curve of continuous treatment was considered, with the maximum effect (100% inhibition) obtained by the slope of avidin control curve, and the percentage of inhibition was defined as the complementary ratio between these two slopes. For calculating the latter parameter, the maximum effect was defined as the measured area between biotin and avidin controls curves. Then, a percentage of reversibility was computed by calculating the complementary ratio between the area separating the continuous and discontinuous treatment curves, normalized by the maximum effect.

### Retrospective *in silico* analysis of different classes of inhibitors

Compound annotations were retrieved from the MeSH (Medical Subject Headings) database, and drugs were clustered in functional groups: protein synthesis inhibitors, DNA synthesis inhibitors, poly (ADP-ribose) polymerase (PARP) inhibitors, and antimetabolites. Each of these clusters was analyzed for its enrichment in high ICD score by means of a Kolmogorov–Smirnov test against the entire compound population.

### Statistical analyses

*In vitro* data are presented as means $\pm$ SD of three experimental replicates if one representative experiment among at least three independent ones is depicted or as means $\pm$ SEM of three independent experiments. *Ex vivo* data are depicted as dot plots with mean $\pm$ SD or as barcharts with mean of $\pm$ SEM (when percentages of positive cells are depicted). Statistical analyses were performed using the freely available software R (https://www.r-project.org) or Excel. One-sided unpaired Student's *t*-test was used to compare

parametric data of different conditions to a control, using the *t*-test function from the *stats* R package or the T.TEST function of Excel. A pairwise multiple comparison test with a Benjamin–Hochberg correction was used to compare each condition to another in a dataset, using the *pairwise.t.test* function from the *stats* R package. A Kolmogorov–Smirnov test was used to compare distributions using the *ks.test* function from the *stats* R package. Pearson correlations were performed, using the *ggscatter* function from the *ggpubr* R package. *In vivo*. Statistical analysis was performed with the TumGrowth software package (Enot *et al*, 2018) freely available at https://github.com/kroemerlab. A type II ANOVA test was used for tumor growth and a log-rank test for mice survival at 1.8 cm². For all tests, the statistical significance levels are shown with stars or hashes: */$^{\#}P < 0.05$, **/$^{\#\#}P < 0.01$, ***/$^{\#\#\#}P < 0.001$. Statistical tests and *P*-values are detailed in Appendix Table S5.

## Data availability

The $GI_{50}$ for a panel of 52,578 compounds was retrieved from the National Cancer Institute website (https://dtp.cancer.gov/databases_tools/bulk_data.htm). Valid PubChem CID was retrieved from PubChem identifier exchange service (https://pubchem.ncbi.nlm.nih.gov/idexchange/idexchange.cgi), and related structure data files (sdf) were obtained from Pubchem. ICD prediction scores were calculated using *ICDPred* available at https://github.com/kroemerlab/ICDpred.

**Expanded View** for this article is available online.

## Acknowledgements
pSpCas9(BB)-2A-GFP (PX458) was a gift from Feng Zhang (Addgene plasmid #48138; http://n2t.net/addgene:48138; RRID:Addgene_48138); Topotecan (609699), 7-Hydroxystausporine (72271), becatecarin (101524), 5-fluorodeoxy-cytidine (515328), and RH-1 (394347) were kindly provided by the NCI. G.K. is supported by the Ligue contre le Cancer (équipe labellisée); Agence National de la Recherche (ANR)—Projets blancs; ANR under the frame of E-Rare-2, the ERA-Net for Research on Rare Diseases; Association pour la recherche sur le cancer (ARC); Cancéropôle Ile-de-France; Chancellerie des universités de Paris (Legs Poix), Fondation pour la Recherche Médicale (FRM); a donation by Elior; European Research Area Network on Cardiovascular Diseases (ERA-CVD, MINO-TAUR); Gustave Roussy Odyssea, the European Union Horizon 2020 Project Oncobiome; Fondation Carrefour; High-end Foreign Expert Program in China (GDW20171100085 and GDW20181100051), Institut National du Cancer (INCa); Inserm (HTE); Institut Universitaire de France; LeDucq Foundation; the LabEx Immuno-Oncology; the RHU Torino Lumière; the Seerave Foundation; the SIRIC Stratified Oncology Cell DNA Repair and Tumor Immune Elimination (SOCRATE); and the SIRIC Cancer Research and Personalized Medicine (CARPEM). J.H. was supported by the foundation Philanthropia. W.X. receives support by the China scholarship council. L.B. is supported by BMS. F.I. benefited from GRISOLIAP/2016/015 grant, from the Regional Ministry for Education in Valencia, and secondly the BEFPI/2018/030 grant, from Regional Ministry for Education in Valencia-European Social Fund. S.L. was supported by Fondation pour la recherche médicale (FRM - FDT201805005722).

## Author contributions
JH performed most experimental procedure, analysis, and preparation of the manuscript. AS performed *in silico* screening and data analysis (for $IC_{60}$ determination, CALR and HMGB1 videos, colocalization assays, and Rush assays) and helped for routine acquisition and analysis of data. GC was in charge of the phagocytosis assays and helped with the construction of the eIF2αS51A cell line. WX generated BMDCs and contributed to assessment of DCs activation markers. FL designed the strategy for constructing the eIF2αS51A cell line. FI contributed to *in vitro* assessment of ICD hallmarks and ER stress markers. LB initiated this project. JPo participated in the design of the immune infiltrate experiment as well as data interpretation. SL and JPa helped with design, experimental procedure, and analysis of the immune infiltrate. ML analyzed tumor slides of xenograft experiments. HT and LZ participated in the conception of the project. OK generated figures and edited the manuscript. GK conceived and directed the project, and wrote the manuscript.

## Conflict of interest
G.K. and O.K. are cofounders of Samsara Therapeutics.

## For more information
www.kroemerlab.com; https://github.com/kroemerlab.

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
