## [Review Process File · EMBO Molecular Medicine]

Inhibition of transcription by dactinomycin reveals a new characteristic of immunogenic cell stress

Juliette Humeau, Allan Sauvat, Giulia Cerrato, Wei Xie, Friedemann Loos, Francesca Iannantuoni, Lucillia Bezu, Sarah Lévesque, Juliette Paillet, Jonathan Pol, Marion Leduc, Laurence Zitvogel, Hugues de Thé, Oliver Kepp, Guido Kroemer

Review timeline:	Submission date:	16th Oct 2019
	Editorial Decision:	7th Nov 2019
	Revision received:	10th Feb 2020
	Editorial Decision:	5th Mar 2020
	Revision received:	25th Mar 2020
	Accepted:	30th Mar 2020

Editor: Céline Carret

Transaction Report:

1st Editorial Decision
2019

7th Nov

Thank you for the submission of your manuscript to EMBO Molecular Medicine. We have now heard back from the two referees whom we asked to evaluate your manuscript.

You will see that their comments are overall supportive. Still, I would like to encourage you to carefully address all comments mentioned by referee 1. Indeed, we believe that a more thorough data analysis in vivo and the addition of clinical correlates would improve the translational and clinical aspect of the paper which is an important point for our scope.

We would therefore welcome the submission of a revised version within three months for further consideration and would like to encourage you to address all the criticisms raised as suggested to improve conclusiveness and clarity. Please note that EMBO Molecular Medicine strongly supports a single round of revision and that, as acceptance or rejection of the manuscript will depend on another round of review, your responses should be as complete as possible.

***** Reviewer's comments *****

Referee #1 (Comments on Novelty/Model System for Author):

The manuscript describes findings that could be potentially translated into the clinical setting. There is a increasing need for data that may help improve current immunotherapy treatments in cancer.

Referee #1 (Remarks for Author):

This is an original study that continues a series of manuscripts from the same principal investigators which have introduced the concept of immunogenic chemotherapy and given major insights into the functional relationship between dying cancer cells and their microenvironment. Here, by using

computational prediction analysis, they identify and select dactinomycin (DACT), a well-known anticancer agent, as an immunogenic cell death (ICD) inducer. They show that DACT treatment is accompanied by all known biochemical and phenotypic hallmarks of ICD (i.e., calreticulin exposure, type I interferon signalling, ATP and HMGB1 release) downstream of endoplasmic reticulum stress response and eIF2 α phosphorylation. These data are validated and supported in in vivo and ex vivo mouse models. Indeed, cancer cells hit with DACT are well taken up by dendritic cells (DCs). Moreover, DACT, either alone or in combination with non ICD inducers (as cisplatin), is much more efficient in the context of an intact immune system. In particular, cognate immune effectors such as CD8 and CD4 T cells are required for an optimal therapeutic success, and the combination with anti-PD-1 blockade lead to long-term tumor growth control or even cure of disease. The notion that DACT is a well-known DNA intercalator and RNA transcription inhibitor, gives to the authors the rationale to investigate whether and find out that inhibition of RNA synthesis is a common feature of immunogenic chemotherapeutics and thus represents yet another hallmark of cancer ICD. This is a very attractive conclusion and could benefit from further clarification and evaluation. The following points are suggested.

- Ex vivo analysis of CD8 T cell activation from MCA205 bearing mice following DACT treatment is not fully convincing. IFN- γ production in DACT group is not significantly higher than Ctr group, which can be at least in part related to CD8 T cell exhaustion and can explain why in this model DACT alone does not work. It would be useful to analyse CD8 T cell state and function either in other tumor models (e.g., WEHI 164) or in MCA205-derived tumors after co-treatment with PD-1 inhibitors.

- Adoptive cell therapy with CD8 T cells from DACT "responder mice" in nu/nu mice would be another compelling option.

- In vitro pulse of cancer cells with DACT significantly enhance calreticulin exposure, DC-mediated uptake of apoptotic bodies and likely tumor antigen cross-presentation. Perhaps the analysis of CD8 T cell activation following co-culture with DCs that have phagocytized cancer cell apoptotic bodies would give a global and more complete view of DACT-induced ICD.

Phagocytosis experimental Ctr at 4{degree sign}C should be shown to rule out the possibility of a mere juxtaposition of DCs with cancer cells instead of an effective taken-up.

- Authors only analyse preclinical settings. The retrospective analysis on publicly available databases of immune infiltrate in patients treated with DACT could help link the conclusion from experimental model back to the patients thereby strengthening the current study.

- Statistics is not uniform, for some experiment SD is reported, for others SEM. Please check.

- Please review few typos (ligand>ligands pag.4; Fig. EVA,K>Fig. EV3A,K pag.8; ionimycin>ionomycin pag.9; fibroscaroma>fibrosarcoma pag.9; crizotinib>crizotinib pag.15).

Referee #2 (Comments on Novelty/Model System for Author):

The in vitro and in vivo models used are appropriate and consistent with the aims of this work. Results are, therefore, consistent.

Referee #2 (Remarks for Author):

In this manuscript, Humeau and colleagues used artificial intelligence to identify anticancer agents that were predicted to induce ICD. They found DACT as a compound with high 'ICD score' and verified this result by both in vitro and in vivo assays. Finally, since DACT is a potent transcriptional/translational inhibitor, they verified whether this feature might represent an ICD requirement.

They have elegantly and convincingly demonstrated this hypothesis by significantly increasing our knowledge of the real impact of ICD in the treatment of neoplasms and provide us with new parameters to evaluate, in a predictive manner, the potential use of new and 'old' antineoplastic compounds.

Therefore, I recommend the manuscript for publication in EMBO Molecular Medicine.

General comment by Reviewer #1: The manuscript describes findings that could be potentially translated into the clinical setting. There is an increasing need for data that may help improve current immunotherapy treatments in cancer.

This is an original study that continues a series of manuscripts from the same principal investigators which have introduced the concept of immunogenic chemotherapy and given major insights into the functional relationship between dying cancer cells and their microenvironment. Here, by using computational prediction analysis, they identify and select dactinomycin (DACT), a well-known anticancer agent, as an immunogenic cell death (ICD) inducer. They show that DACT treatment is accompanied by all known biochemical and phenotypic hallmarks of ICD (i.e., calreticulin exposure, type I interferon signalling, ATP and HMGB1 release) downstream of endoplasmic reticulum stress response and eIF2 α phosphorylation. These data are validated and supported in *in vivo* and *ex vivo* mouse models. Indeed, cancer cells hit with DACT are well taken up by dendritic cells (DCs). Moreover, DACT, either alone or in combination with non ICD inducers (as cisplatin), is much more efficient in the context of an intact immune system. In particular, cognate immune effectors such as CD8 and CD4 T cells are required for an optimal therapeutic success, and the combination with anti-PD-1 blockade leads to long-term tumor growth control or even cure of disease. The notion that DACT is a well-known DNA intercalator and RNA transcription inhibitor, gives to the authors the rationale to investigate whether and find out that inhibition of RNA synthesis is a common feature of immunogenic chemotherapeutics and thus represents yet another hallmark of cancer ICD. This is a very attractive conclusion and could benefit from further clarification and evaluation.

Our response: We appreciate the encouraging comments by reviewer #1.

Point 1 raised by Reviewer #1: *Ex vivo* analysis of CD8 T cell activation from MCA205 bearing mice following DACT treatment is not fully convincing. IFN- γ production in DACT group is not significantly higher than Ctr group, which can be at least in part related to CD8 T cell exhaustion and can explain why in this model DACT alone does not work. It would be useful to analyse CD8 T cell state and function either in other tumor models (e.g., WEHI 164) or in MCA205-derived tumors after co-treatment with PD-1 inhibitors. Adoptive cell therapy with CD8 T cells from DACT "responder mice" in *nu/nu* mice would be another compelling option.

Our response: We thank the reviewer for this comment. Since the tumors are relatively small after DACT treatment in the WEHI 164 model, the number of tumor-infiltrating lymphocytes is low, rendering it difficult to perform adoptive transfer experiments. However, we have addressed the question about interferon-gamma production by means of quantitative RT-PCR (new Fig. 6). The tumors from DACT-treated mice contained more mRNA coding for interferon-gamma than control tumors from untreated mice. These results were obtained 9 days after DACT treatment, when due to the small tumor size, cytofluorometric analyses of the tumor-infiltrating lymphocytes is problematic. However, to obtain information on the contribution of the immune system, we depleted T lymphocytes (with CD4 and CD8-specific antibodies), showing that this maneuver abolished the therapeutic effects of DACT. We performed additional experiments in which we injected a neutralizing interferon-gamma specific antibody, showing that this also impaired tumor growth reduction by DACT (new Fig. 6).

Finally, we rechallenged mice that had been cured from established WEHI164 sarcoma by DACT-based chemotherapy with the same cells (WEHI164 cells injected into the opposite flank), finding that the mice had developed a protective immune response that

precluded the growth of the sarcoma cells. As a control, such cells rapidly formed tumors in naïve mice, as determined in the same experiment (new Fig. 5I-K).

Point 2 raised by Reviewer #1: In vitro pulse of cancer cells with DACT significantly enhance calreticulin exposure, DC-mediated uptake of apoptotic bodies and likely tumor antigen cross-presentation. Perhaps the analysis of CD8 T cell activation following co-culture with DCs that have phagocytized cancer cell apoptotic bodies would give a global and more complete view of DACT-induced ICD.

Our response: To assess if DACT-treated tumors induce activation and maturation of DCs, we co-cultured DACT-treated MCA205 cells with BMDCs during 24 h and measured the percentage of co-stimulatory molecule CD86 and of MHC class II positive CD11c cells. DACT was indeed able to increase these markers of DC activation (Fig. 4A-D, Fig. S3).

Point 3 raised by Reviewer #1: Phagocytosis experimental Ctr at 4°C should be shown to rule out the possibility of a mere juxtaposition of DCs with cancer cells instead of an effective taken-up.

Our response: As requested, the phagocytosis experiment has been repeated including additional controls. As expected, DACT-treated tumor cells were significantly less phagocytosed at 4°C as compared to 37°C standard environmental conditions. This new data is included in the revised version of the manuscript (Fig. 4B, Fig. S2).

Point 4 raised by Reviewer #1: Authors only analyse preclinical settings. The retrospective analysis on publicly available databases of immune infiltrate in patients treated with DACT could help link the conclusion from experimental model back to the patients thereby strengthening the current study.

Our response: We have done our best to locate databases describing the molecular properties of the immune infiltrate in clinical tumor specimens from patients under DACT treatment. Commensurate with the fact that DACT is only rarely used in modern oncology (mostly in the context of rare pediatric tumors), we were unable to find this information.

Minor point 5 raised by Reviewer #1: Statistics is not uniform, for some experiment SD is reported, for others SEM. Please check.

Our response: We have reformulated the statistic evaluation part in the Material and Methods section and pointed out the statistical methods and the basis of their usage: SD is used to show variation of experimental replicates when data come from one representative experiment. SEM is used to show variation of *ex vivo* data or when the mean of at least three independent experiment is shown.

Minor point 6 raised by Reviewer #1: Please review few typos (ligand>ligands pag.4; Fig. EVA,K>Fig. EV3A,K pag.8; ionimycin>ionomycin pag.9; fibroscaroma>fibrosarcoma pag.9; crizotinib>crizotinib pag.15).

Our response: We apologize for these oversights. The manuscript has been re-edited and orthographic flaws have been addressed.

General comment by Reviewer #2: The in vitro and in vivo models used are appropriate and consistent with the aims of this work. Results are, therefore, consistent.

In this manuscript, Humeau and colleagues used artificial intelligence to identify anticancer agents that were predicted to induce ICD. They found DACT as a compound with high 'ICD score' and verified this result by both in vitro and in vivo assays. Finally, since DACT is a potent transcriptional/translational inhibitor, they verified whether this feature might represent an ICD requirement.

They have elegantly and convincingly demonstrated this hypothesis by significantly increasing our knowledge of the real impact of ICD in the treatment of neoplasms and provide us with new parameters to evaluate, in a predictive manner, the potential use of new and 'old' antineoplastic compounds.

Therefore, I recommend the manuscript for publication in EMBO Molecular Medicine.

Our response: We are very grateful to reviewer #2 for his encouraging support of our work.

2nd Editorial Decision

5th Mar 2020

Thank you for the submission of your revised manuscript to EMBO Molecular Medicine. We have now received the enclosed reports from the referees that were asked to re-assess it. As you will see the reviewers are now supportive of publication and I am pleased to inform you that we will be able to accept your manuscript pending the following final amendments.

***** Reviewer's comments *****

Referee #1 (Remarks for Author):

The authors have addressed all my suggestions.

Referee #2 (Remarks for Author):

The authors produced new data to answer the reviewer's criticisms and included them in the revised manuscript. In my opinion, they cleared any point raised by the reviewer, and the new data also contribute to increasing the overall quality of the manuscript. I, therefore, recommend the manuscript for publication in the present form.

2nd Revision - authors' response

25th Mar 2020

The authors performed the requested editorial changes.

Corresponding Author Name: Oliver Kepp, Guido Kroemer

Journal Submitted to: EMBO Mol Med

Manuscript Number: EMM-2019-11622-V2